# Effects of group entitativity on young English-speaking children's interpretation of inclusive *We*

Jared Vasil[1]*, Camryn Capoot[1], Michael Tomasello[1,2]

1 Duke University, Durham, North Carolina, United States of America, 2 Max Planck Institute of Evolutionary Anthropology, Leipzig, Germany

* jared.vasil@duke.edu

**Data Availability Statement:** All data and code for the studies are freely available at: https://osf.io/5cs9p/.

**Funding:** MT: Jacobs Foundation Klaus J Jacobs Research Prize (https://jacobsfoundation.org/

## Abstract

The pronoun *we* can be used to refer to various collections of people depending on various pragmatic factors. This article reports the results of two online experiments that investigated children's interpretation of inclusive *we*, in which the child-listener is part of the intended referent of *we*. 128 2- and 4-year-olds collaborated with three partners in a coloring task. Before they played together, one partner informed participants that, e.g., "we can color!" Participants had their own markers and had to choose to how many partners to distribute (virtual) markers. In the first experiment, the partners appeared more like an aggregation of individuals than a collaborative group. The second experiment flipped this so that the partners appeared more like a collaborative group. Contrary to expectations, there was relatively little evidence for development in children's interpretation of *we*. Additionally, participants did not sharply distinguish their interpretations of *we* from those of *we both* or *we all*. Rather, participants were more likely to choose group interpretations when contextual cues indicated that their partners were a collaborative group than an aggregation of individuals. Interestingly, this interpretational distinction was sharpest for the pragmatically ambiguous *we*, compared to the relatively unambiguous *we both* and *we all*. These results are informative about the kinds of cues that shape young children's interpretation of pragmatically ambiguous pronominal reference.

## Introduction

Human psychology is characteristically "groupminded." Adults uphold commitments, enforce norms, and share fairly. Groupmindedness relies on the ability to conceive of social groups and to leverage the group's perspective to guide behavior. Groupminded thinking has an ontogenetic trajectory. While children younger than 3 years of age jointly engage in coordinated social play and reengage disengaged partners [1, 2], only after they turn 3 do children begin to uphold commitments [3], enforce norms [4], and share fairly [5]. That is, children younger than 3 years can only easily leverage a dyadic perspective on collaboration. In contrast, children older than 3 can, also, easily leverage a group perspective on collaboration, too (for

klaus-j-jacobs-research-prize/) The funders had no role in study design, data collection and analysis, decision to publish, or preparation of the manuscript.

**Competing interests:** The authors have declared that no competing interests exist.

distinctions between dyads and groups, see [6]). Three years of age is thus a watershed moment in the ontogeny of mature human psychology [7]. At this age, the kinds of easily conceptualizable social entities changes qualitatively–after 3 years, children can easily conceive of dyads or groups.

Does the groupminded shift at 3 years of age influence language development? To answer this question, research has focused on linguistic reference. Successful reference requires that listeners form inferences about speakers' intended referents, given linguistic form. How listeners solve this problem depends, in part, on the entities that they can conceptualize [8, 9]. For example, if the groupminded shift expands the set of conceptualizable social entities (thus, conceptualizable referents), then children older than 3 years may form three-person "group interpretations" of plural person reference more easily than children younger than 3. Instead, those younger children may favor 2-person "dyadic interpretations."

Indeed, research on English-speaking children's comprehension of *we* has supported this hypothesis. Use of the word *we* only partially specifies speaker referential intentions. While use of *we* indicates that speakers intend to refer to themselves and to other(s), it is ambiguous whether the latter referential intention includes only one individual, or two or more individuals. Note that this is a kind of "pragmatic" ambiguity in referential intentions (see also [10]), and not semantic ambiguity (i.e., as "ambiguity" is understood to pertain to truth conditions). Nonetheless, resolution of this ambiguity is psychologically important for listeners because the former referential intention warrants a dyadic interpretation, whereas the latter warrants a group interpretation.

[11] investigated how 2- and 4-year-olds resolve this pragmatic ambiguity in use of *we*. Those authors found that 4-year-olds systematically formed dyadic or group interpretations of *we*, depending on the spatial position of a speaker relative to two other individuals. In contrast, 2-year-olds favored dyadic interpretations. Importantly, 2- and 4-year-olds displayed an adult-like understanding of the first-person semantics of *we* by reliably including the speaker as one of the individuals referred to by use of *we* (number was controlled, which precluded investigating children's understanding of the plural semantics of *we*). Consequently, [11] interpreted this developmental shift in referential interpretation as evidence that a groupminded shift at 3 years expands the set of conceptualizable social entities. Corroboratory results for referential production are reported by [12].

However, there was a limitation of the study of [11]. Participants in that study were excluded from (i.e., were third parties to) the intended referent of *we*. Rather, in that study, speakers referred only to themselves and other (i.e., non-participant) individuals. This limitation is important because children may have more experience interpreting *we* when they are included in the intended referent of *we*. That is, children may have less experience with the "exclusive *we*" used in [11] than they do with the "inclusive *we*." Indeed, the fact that young children hear *we* infrequently relative to most other pronouns arguably makes this limitation more pressing. For example, [12] found that *we* and other first-person plural personal, possessive, and reflexive pronouns constitute about 6% of mothers' spontaneous pronouns and occur in only about 3% of mothers' spontaneous utterances. The present studies remedied this limitation by examining children's interpretations of inclusive *we*, in which children were part of the intended referent of *we*.

Some languages formally mark the "clusivity" of first-person plural pronouns [13]. However, English does not. For instance, the Mandarin Chinese first-person plural form *wámen* indicates a listener-inclusive intended referent, whereas the listener-exclusive form is *zánmen*. In contrast, English requires the use of multiword constructions to mark intended clusivity, e.g., *we but not you*, *you and I*, *us two*, etc.; or else, listeners rely on contextual cues (e.g., joint

commitment) to appropriately disambiguate intended clusivity. The present experiments leveraged the latter option.

To do this, Study 1 implemented a novel, virtual method for investigating children's interpretation of ambiguous reference during collaboration. Next, Study 2 built on Study 1 by increasing the entitativity of participants' partners [14], that is, their appearance as a group working together towards a shared goal. In both studies, participants collaborated with three partners. One of the partners used *we*, *we both*, or *we all* to refer to the collaborators. Participants had to infer the speaker's intended referent. First, developmental change in was predicted in the *we* condition, only. Specifically, 2-year-olds were predicted interpret *we* like *we both* (i.e., dyadic interpretations), whereas 4-year-olds were predicted to interpret *we* like *we all* (i.e., group interpretations). Second, it was predicted that increasing the group entitativity of participants' partners would increase participants' tendency to form group interpretations of *we*, only, and not *we both* or *we all*. This prediction was made because *we* is, as noted above, formally ambiguous with respect to speaker referential intentions. Thus, it was predicted that altering contextual cues to increase the group entitativity of participants' partners would pull for participants to assign group interpretations to *we*. However, because *we all* and *we both* are not formally ambiguous in this way, no such effect was expected for those forms. Analyses presented in Study 1 and Study 2 investigated the first prediction. Analyses presented in the Comparison of Study 1 and Study 2 investigated the second prediction. Both studies also investigated effects of sex on interpretation of *we*. Previous studies of pronoun development suggest that males may form group interpretations of *we* more often than females [11]. Complete code and data used in the analyses reported in this article are freely available at OSF (https://osf.io/5cs9p/?view_only=e3c045f73d2b4ef08897b4f4718a9cd0). The experimenter scripts used in the procedures of the present studies are available at the link, too.

## Study 1

### Methods

**Participants.** There were 71 participants. The final sample included 64 participants, 32 2-year-olds (*median* = 2.54 years, *range* = 2.25–2.74 years, 18 males) and 32 4-year-olds (*median* = 4.60 years, *range* = 4.29–4.76 years, 13 males). Seven participants had all three trials excluded (*N* = 3 2-year-olds, *N* = 4 4-year-olds; exclusion criteria below). Caregivers predominantly indicated that households made more than $100,000 per year (*N* = 42) or between $60,000-$100,000 per year (*N* = 17). Caregivers predominantly identified participants as White (*N* = 49) or Biracial-Asian/White (*N* = 7). Caregivers received a $10 Amazon gift card. Participants received a certificate. Study design and procedure were approved by the Duke University Campus Institutional Review Board (protocol 2021–0604). Participants were sampled from October 8, 2021 to March 16, 2022. Written informed consent was obtained from participants' parents or legal guardian prior to participation in the procedure. Participants were minors and were typically developing (per caregiver statement).

**Design.** A three-level independent variable was manipulated within-subjects. The levels of this variable were labelled *we*, *we both*, and *we all*. The *we* condition was presented first. The order of presentation of the *we both* and *we all* conditions was counterbalanced between participants. This was intended to preempt carryover effects (e.g., from *we all* to *we*). There were seven warmup trials and three test trials. The order of presentation of the warmup trials was counterbalanced between participants (individuals first or groups first; see below). The procedure was remotely moderated [15] via Zoom [16]. Specifically, the experimenter (E) interacted with participants while the latter's "interactions" with their puppet partners were, in fact, with pre-recorded videos surreptitiously controlled by E and contingent on participant behavior.

**Materials.** Three hand puppets (a lion, a giraffe, and a monkey), several 8 x 11-inch sheets of paper, a set of monochromatic blocks, a set of coloring markers, and a small table. A male native speaker of English voiced lion and two females voiced giraffe and monkey, respectively. No effect of puppet voice on participants' responses was found, so analyses collapse across this variable.

**Procedure.** E greeted caregivers and participants on Zoom before initiating a Zoom Setup Phase. During the Zoom Setup Phase, E showed a PowerPoint to caregivers that illustrated how their screen should appear (i.e., full, self-view hidden, appropriate volume). Caregivers were asked to remain silent during the procedure, except to refocus participants' attention onscreen if it wavered.

The Zoom Setup Phase was followed by a Warmup Phase. In the Warmup Phase, E told participants that they will "meet some new friends." Then, E played a video in which each of three puppets was presented individually, in sequence, onscreen. Each puppet greeted participants, in turn, by saying "Hi, I'm [e.g.] Lion!" followed by E pausing the video to say, "Hi Lion!" After greeting each partner, participants viewed a screen in which all three partners were aligned behind a table, roughly equidistant from one another. Then, E said "Hey [participant's name], I think that Lion, Giraffe, and Monkey want to play with you, [participant's name]!" E then stated that they have blocks that "you" (i.e., the puppets) can play with. Several images of colorful wooden blocks then appeared onscreen below the frame. Next, E asked participants whether they have their blocks. (Prior to participation, E asked caregivers to bring a set of blocks.) Participants showed or indicated the presence of their blocks. Next, E said "I think [e.g.] Lion wants to play with you! [Participant's name], who do you think I should give blocks to?" Participants could respond freely. During the warmup, E always 'gave' the blocks to the puppet(s) who E suggested wanted to play with participants, regardless of how participants responded. Specifically, E said "Hm. . . I think I will give the blocks to [e.g.] Lion" before 'giving' the blocks to Lion by dragging the image of the blocks immediately in front of the Lion puppet. E then asked participants if they were ready "to start playing with" the partner(s) who possessed blocks. Once participants agreed, E played the appropriate video of the puppet(s) playing with the blocks. This video lasted approximately 10s and depicted the puppet(s) stacking a real set of blocks that resembled those depicted in the pictured blocks. Puppet(s) to whom E did not distribute blocks remained onscreen, standing still while the other puppet(s) built with the blocks. Participants were free to play with their own blocks during this time. Once the video completed, E indicated that another puppet(s) wanted to play with the participant, e.g., Giraffe. The above was repeated for each combination of puppets. That is, participants played blocks with each combination of puppets. This resulted in seven warmup trials. Counterbalanced between participants was order of warmup presentation, individuals first or groups first. The individuals first order began with participants playing with each of the three puppets, in turn, before playing with each puppet dyad, in turn, (e.g., Lion and Giraffe, etc.) and then all three puppets at once. The groups first order proceeded in reverse order. E concluded the Warmup Phase by displaying a screen in which the blocks were absent. E noted the absence of the blocks and suggested that participants color with their partners, next. In sum, the Warmup Phase introduced participants to their puppet partners and familiarized them with the task employed in the Test Phase. Specifically, the warmup familiarized participants with the structure of the task (e.g., that E could move the images onscreen to 'give' the toy) and demonstrated that every puppet, and combination of puppets, could receive the toy.

The Test Phase followed the Warmup Phase. E began by saying that E had to briefly leave but that, while E was gone, participants could talk to the puppets because "I think they want to color with you!" E then left the camera's view while surreptitiously controlling the videos. There were three test trials. Each test trial featured a different speaker puppet, e.g., on Test

Trial 1 the speaker puppet was Lion, on Test Trial 2 Giraffe, etc. (Fig 1). The speaker puppet was the puppet with whom participants 'spoke' during the trial. Specifically, on Test Trial 1, e.g., Giraffe said "Do you like to color too? [*E pauses video, participant responds*] Great! I love to color. Coloring is so much fun. Do you want to color? Does coloring sound fun to you?" [*E pauses video, participant responds*]. If participants did not respond to a question asked by the speaker puppet, E repeated the question to participants by saying that, e.g., "Giraffe asked if coloring still sounds fun to you. Does coloring still sound fun to you?" In the *we condition*, the speaker puppet said "Great! **We** can color! I think **we** can color! **We** can color together!" (The bolded terms were replaced with **we all** or **we both** in the *we all condition* and *we both condition*, respectively.) Next, the speaker puppet remarked on how there were no markers present by asserting "Oh no, wait! There are no markers! There's no coloring without markers! I really wish there were markers, but there are no markers. If there were markers, then **we** could color together!" While the speaker puppet said this, E reappeared onscreen. Once the speaker puppet finished speaking, E pointed out that "I have some markers!" to color with. Then, three images of markers appeared onscreen (Fig 1), identically to how the images of blocks appeared in the Warmup Phase. The speaker puppet reacted happily to the appearance of the markers. The speaker puppet then asked participants whether they had their markers, too: "Oh, hey, wait, did you bring your markers, too?" [*E pauses video, participant responds*] "Great! You have your markers, too. Now, **we** can color! Now, **we** can color together!" With the markers displayed onscreen, E then asked participants "who does [e.g.] Giraffe want to color with? Who does Giraffe want to give markers to?" Participants responded freely. E clicked and dragged the image(s) of the markers to the puppet(s) who participants indicated. Once participants finished responding, the speaker puppet said "Thanks! Now, time to color!" Then, E chose the video that depicted the appropriate puppet(s) drawing with markers (i.e., those indicated by participants). E echoed that it was "Time to color!" and played the appropriate video. The video lasted approximately 10s and depicted the appropriate puppet(s) drawing on a sheet of paper placed in front of each puppet (Fig 1). Puppet(s) to whom participants did not distribute markers remained onscreen, standing still while the other puppet(s) colored. Participants colored freely. After 10s, E said "Okay, time's up, I need my markers back!" just before the speaker puppet said "Okay, here's your markers back" at which point the puppet(s) in the video placed

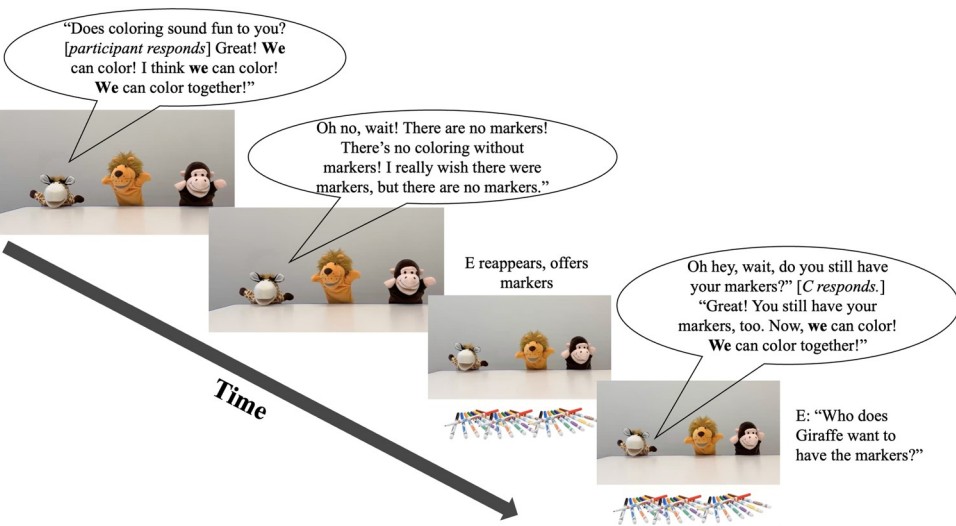

**Fig 1. Timeline of a test phase trial.**

their markers down (i.e., so that the speaker puppet's speech and act of putting the markers down appeared contingent on E's request). The second and third trials of the Test Phase were identical to the above, save for the use of *we all* or *we both* instead of *we*. The second and third trials were introduced by a new speaker puppet commenting to the participant that "Coloring still seems like so much fun! Do you still want to color?"

**Coding.**   To be included in the final sample, participants' caregivers had to indicate that the participant's primary language was English. Trial-level data was excluded due to sibling or caregiver interference, ambiguous/unclear participant response, or if participants did not include the speaker puppet in their response (in Test Phase trials). This latter follows from the fact that referents of uses of *we*, *we all*, or *we both* must include speakers. Participants could indicate their responses linguistically or nonlinguistically, e.g., via pointing. When possible, nonlinguistic responses were clarified by caregivers. Otherwise, nonlinguistic responses were excluded when ambiguous. In the final analysis, 153 trials were included ($N$ = 64 participants) and 39 trials were excluded ($N$ = 26 of the 64 participants). Of the 153 included trials, 38 participants contributed three trials (114 trials), 13 contributed two trials (26 trials), and 13 contributed one trial (13 trials). All data was initially coded by a research assistant who was blind to the hypotheses of the study. Reliability coding was performed on a random 25% of participants' responses ($N$ = 48 trials) by the first author. Responses were coded according to whether participants chose to give markers to 1, 2, or 3 puppets, and whether one of those puppets was the speaker puppet. Interrater reliability for the number of puppets to whom participants chose to distribute markers was excellent, $\kappa$ = 0.96, and agreement was excellent, 96% of trials.

**Data analysis plan.**   Hierarchical Bayesian models were fitted to the data [17] via the Stan [18] front-end interface brms [19]. The dependent variable was dichotomous ("dyadic interpretation" or "group interpretation"). The dyadic interpretation corresponded to participants having selected only the speaker puppet to receive the markers. In contrast, the group interpretation corresponded to participants having selected the speaker puppet and at least one of the other two puppets to receive the markers. Thus, there were two group interpretations and one dyadic interpretation that participants could choose.

*Main analysis*. Two models were fitted and analyzed in the main analysis. A control variables model included first-order predictors of order of presentation of conditions (*we all* second, *we both* second), test trial (first, second, third), speaker puppet (giraffe, monkey, lion), and participant sex (male, female). A main model included the key predictor of condition (*we*, *we all*, *we both*), participant age group (2 years or 4 years), and their interaction. Both models included a random intercept of participant and random slope of condition. The control variables model formula was:

interpretation $\sim$ condition order + test trial + speaker + sex + (condition||participant)

The main model formula was:

interpretation $\sim$ age group $*$ condition + (condition $\|$ participant)

Variance components correlations were not modeled because they equaled zero due to counterbalancing. Control model predictors were sum coded. Main model predictors were treatment coded. The baseline groups in the main model were the *we* condition (Condition) and 2-year-olds (Age). Priors were $\beta_{intercept} \sim t(10,0,1), \beta_{predictors} \sim N(0,0.75), \sigma \sim t(10,0,0.4)$. Posterior parameter estimates were characterized by their 95% highest density (HDI) and proportion of density over positively signed parameter values. For interpretational ease, qualitative assertions about the posterior evidence for positively signed values are provided. "Weak" means that between 10% and 90%, "moderate" between 5% and 10% or 90% and 95%, and "strong" that

less than 5% or more than 95% of posterior mass covers values greater than 0. Stability of posterior estimates was ensured via sensitivity analysis with flatter and peakier fixed effects priors. All R-hats equaled 1.00 and effective sample sizes were adequate (following [17]).

Priors were chosen based on general domain knowledge and an understanding of the response distribution defined by the link function so as to be "weakly informative" about the most likely outcomes. This strategy is considered best practice in Bayesian modeling [e.g., 17, 20–22, provides an introduction] and has been used previously [11, 23].

*Follow-up analyses*. Simpler models were fitted to portions of the dataset investigated in the main analysis. All models reported included the same random effects structure as included in the model reported in the main analysis, or a simplified version thereof, as appropriate. Fixed effects structures are detailed before reporting the fitted model. A first follow-up analysis investigated associations between interpretation and age within conditions; a second investigated the effect of condition on interpretation within ages; and a third investigated interpretations against chance.

## Results

**Data.**   The distribution of participant interpretations is depicted by condition in Fig 2. Two- and 4-year-olds generally favored dyadic over group interpretations. That is, Study 1 participants tended to distribute markers to the speaker puppet, only, and not to the other two puppets. Interestingly, 2-year-olds in the *we all* condition made group interpretations relatively more often than any other condition *x* age pair, with 9 of 20 2-year-olds choosing group

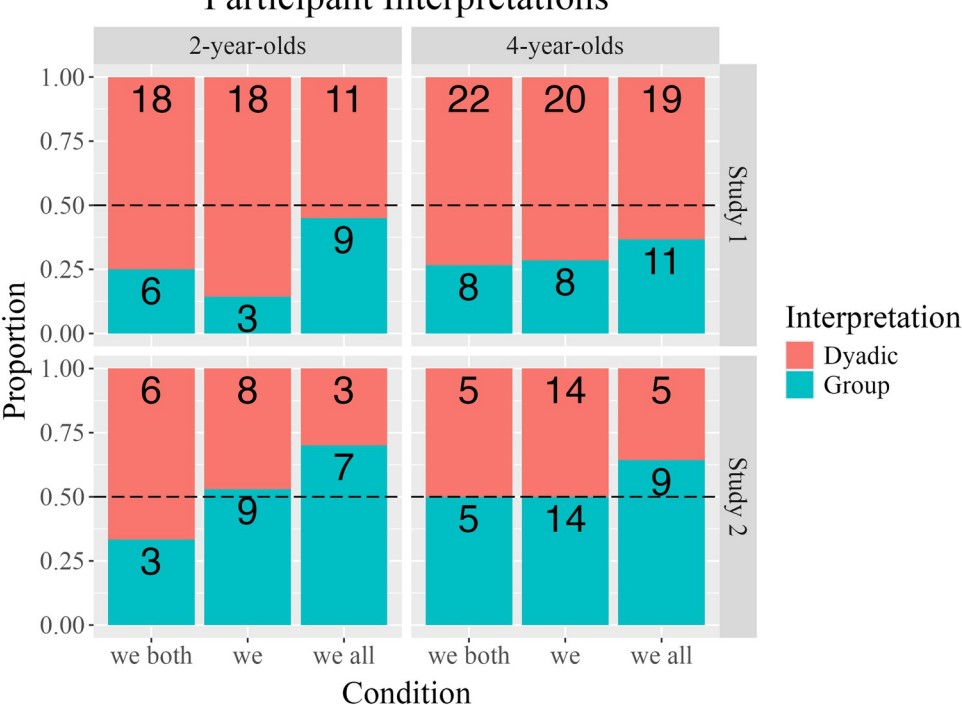

**Fig 2. Trial-level data by condition.** Trial-level distribution of participants' interpretations, by condition, in Study 1 and Study 2 (top and bottom facets, respectively). Study 1 facet displays data for Test Trials 1, 2, 3. Study 2 facet displays data for Test Trials 1 and 2 (because of a reliable association between the DV and the order of the *we both* and *we all* conditions; see Study 2).

interpretations of *we all*. S1 Fig displays the dependent variable as a trichotomous outcome (i.e., participants' distribution of markers to 1, 2, or 3 puppets).

**Main analysis.** The posterior control variables model was assessed, first. Most predictors were unreliably related to interpretations (see S1 Table for posterior parameter estimates). The only exception was a reliable posterior association that suggested that participants chose group interpretations less often when giraffe, rather than Monkey, was the speaker puppet, -0.48, [-1.17,0.19], $\Pr(\beta > 0.00|D) = .08$. There was weak evidence that females chose group interpretations more often than males (Fig 3), 0.11, [-0.64,0.89], $\Pr(\beta > 0.00|D) = .61$. S2 Fig reports participant-level data, by sex.

The posterior main model was assessed next. Developmental change was predicted in interpretation of *we*. Specifically, 4-year-olds were predicted to choose group interpretations more often than 2-year-olds. It was also predicted that 2-year-olds would interpret *we* like *we both*, whereas 4-year-olds would interpret *we* like *we all*.

Based on a model fitted to the data reported in Fig 2, mixed support for these predictions was found. There was strong evidence that 2-year-olds in the *we* condition chose group interpretations below chance levels, -1.70, [-2.89,-0.63], $\Pr(\beta > 0.00|D) = .00$, and 4-year-olds did not reliably diverge from this pattern, 0.16, [-0.93,1.28], $\Pr(\beta > 0.00|D) = .62$. That is, 2- and 4-year-olds favored dyadic interpretations of *we*. The lack of developmental change represented in this latter finding discords with our predictions. Nonetheless, 2-year-olds made group interpretations of *we both* at similarly depressed rates, -0.03, [-1.06,0.97], $\Pr(\beta > 0.00|D) = .48$. This finding accorded with predictions. Taken with the first finding, 2-year-olds favored dyadic interpretations at similar rates in the *we* and *we both* conditions. Moreover, there was some evidence that 2-year-olds chose group interpretations of *we all* more often than of *we*, 0.69, [-0.32,1.68], $\Pr(\beta > 0.00|D) = .91$. This finding accorded with predictions, though was not particularly robust. There was weak evidence that 4-year-olds interpreted *we all* and *we both*, relative to *we*, differently than 2-year-olds, *we all*: -0.01, [-1.15,1.15], $\Pr(\beta > 0.00|D) = .49$; *we both*: 0.02, [-1.11,1.12], $\Pr(\beta > 0.00|D) = .51$. Altogether, 2- and 4-year-olds, in situations like that of the present study, form largely similar interpretations of *we*, *we both*, and *we*

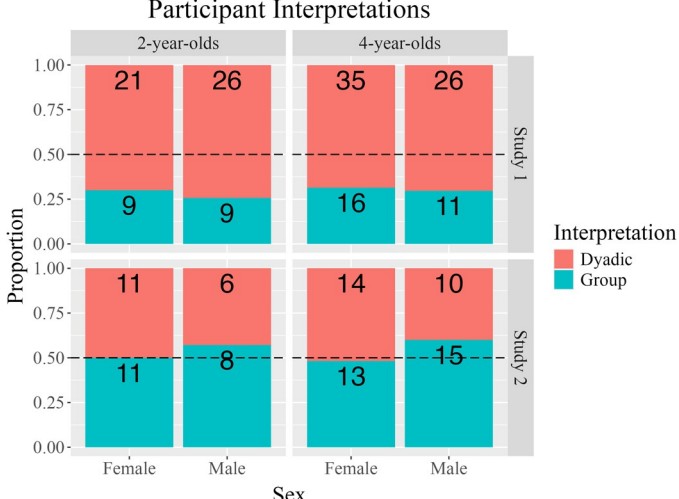

**Fig 3. Trial-level data by sex.** Trial-level distribution of participants' interpretations, by participant sex, in Study 1 and Study 2 (top and bottom facets, respectively). Study 1 facet displays data for Test Trials 1, 2, 3. Study 2 facet displays data for Test Trials 1 and 2 (because of a reliable association between the DV and the order the *we both* and *we all* conditions; see Study 2). S2 Fig depicts participant-level data.

*all*. Specifically, young children tend to interpret these forms referring to dyads, not groups. Interestingly, 2-year-olds' interpretations distinguished between *we* and *we all* more than those of 4-year-olds.

**Follow-up analyses.** Three sets of follow-up analyses were conducted. For readers' ease, only the proportion of posterior mass greater than 0 is reported. Summary statistics of marginal posteriors are reported in S2 Table. A first set of follow-up analyses investigated developmental change by condition. One model was fitted to the data in each condition. Each model included a fixed effect of age group, only. The strongest evidence for developmental change was found in the *we* condition. 81% of posterior samples indicated that 4-year-olds chose group interpretations more often than 2-year-olds, $\Pr(\beta > 0.00|D) = .81$. This pattern accorded with the predictions of the current study but was less robust than anticipated. As expected, there was little evidence for reliable associations with age in the other conditions, *we both* condition: $\Pr(\beta > 0.00|D) = 55$; *we all* condition, $\Pr(\beta > 0.00|D) = .33$. Overall, this first set of follow-up analyses broadly aligns with the predictions of this study. The strongest evidence of developmental change was in participants' interpretation of *we*. Specifically, there was some evidence that suggested 4-year-olds chose group interpretations of *we* more often than 2-year-olds.

A second set of follow-up analyses investigated the effect of condition within age group. One model was fitted to the data in each age. Both models included a fixed effect of condition, only. There was weak evidence that participants chose group interpretations at different rates in the *we* and *we both* conditions, 2-year-olds: $\Pr(\beta_{condition} > 0.00|D) = .55$, 4-year-olds: $\Pr(\beta_{condition} > 0.00|D) = .38$. That is, the distribution of 2-year-olds' interpretations of *we* and *we both* were similar (as predicted), but so were 4-year-olds' (unpredicted). This pattern accords with the findings of the main analysis, above. There was weak evidence that 4-year-olds chose group interpretations at different rates in the *we* and *we all* conditions, $\Pr(\beta_{condition} > 0.00|D) = .73$. Thus, as predicted, 4-year-olds' interpretations of *we* resembled their interpretations of *we all*. In contrast, there was moderate evidence that 2-year-olds chose group interpretations at a higher rate in the *we all* condition than in the *we* condition, $\Pr(\beta_{condition} > 0.00|D) = .92$. Though less robust than expected, this finding accorded with predictions– 2-year-olds chose group interpretations of *we* at lower rates than of *we all*. Overall, this second set of follow-up analyses suggested mixed evidence for the predictions of this study. While 2-year-olds' interpretations of *we* resembled their interpretations of *we both* more than *we all* (as predicted), 4-year-olds' interpretations of *we* were similar across conditions (not predicted).

A third set of follow-up analyses compared the distribution of participants' responses against chance. Three models were fitted to each condition * age group pair (i.e., there were six models in total). Each model only modeled the intercept (baseline) rate of group interpretations. Conceptually, "chance" responses in the referring situation captured in the procedure can be defined in two ways. First, the chance rate of choosing group interpretations was defined as 0.50. This definition stipulated that 3- and 4-person groups were not perceived by participants as distinct interpretations. Under this definition of chance, 2- and 4-year-olds mostly picked group interpretations less often than chance, 2-year-olds: $\Pr(\beta_{we} > 0.00|D) = .00$; $\Pr(\beta_{we\ both} > 0.00|D) = .01$; 4-year-olds: $\Pr(\beta_{we} > 0.00|D) = .02$; $\Pr(\beta_{we\ both} > 0.00|D) = .01$; $\Pr(\beta_{we\ all} > 0.00|D) = .08$. Unexpectedly, the sole exception to this pattern was 2-year-olds in the *we all* condition, who did not reliably choose dyadic over group interpretations, $\Pr(\beta_{we\ all} > .00|D) = .34$. Next, the chance rate of choosing group interpretations was defined as 0.66. This analysis assumed that 3-person and 4-person groups were perceived by participants as distinct interpretations. Under this definition of chance, 2- and 4-year-olds picked group interpretations below chance levels, 2-year-olds: $\Pr(\beta_{we} > 0.69|D) = .00$; $\Pr(\beta_{we\ both} > 0.69|D) = .00$; $\Pr(\beta_{we\ all} > 0.69|D) = .02$; 4-year-olds: $\Pr(\beta_{we} > 0.69|D) = .00$; $\Pr(\beta_{we\ both} > 0.69|D) = .00$; $\Pr(\beta_{we\ all} > 0.69|D) = .00$. Overall, this third set of follow-up analyses suggested that participants

generally favored dyadic interpretations in Study 1. Interestingly, however, 2-year-olds did not reliably favor dyadic or group interpretations of *we all* (while though 4-year-olds favored dyadic interpretations).

In sum, Study 1 participants typically interpreted *we*, *we both*, and *we all* dyadically. The follow-up analyses provided mixed support for the study predictions. Notably, there was some evidence that 4-year-olds made group interpretations of *we*, and not *we both* or *we all*, more often than did 2-year-olds. This key finding accorded with the predictions of the current study.

## Study 1 Discussion

Study 1 investigated children's interpretation of *we*. To do this, participants' behavior was investigated in a context in which collaboration relied upon participants interpreting uses of *we*, *we both*, or *we all* (this was thus an "inclusive" *we*). Based on a hypothesized groupminded shift at 3 years [7], 2-year-olds were predicted to prefer dyadic interpretations of *we*. In contrast, 4-year-olds were predicted to prefer group interpretations of *we*. That is, 2-year-olds' interpretations were predicted to resemble their interpretations of *we both* and 4-year-olds' those of *we all*. However, participants mostly favored dyadic interpretations (one exception was 2-year-olds' interpretation of *we all*, where group interpretations were comparatively frequent). There was little evidence for associations between other predictors and participant interpretations.

What explanation might account for the lack of evidence for the present set of hypotheses? Perhaps the context of the procedure used in Study 1 pulled too strongly for dyadic interpretations, resulting in a floor effect. If this were the case, a context that pulls more for group interpretations may increase the size of observed behavioral differences between conditions.

## Study 2

Study 2 implemented a procedure that was essentially identical to that of Study 1. Changes targeted the group entitativity of participants' puppet partners. These were enacted by modifying contextual cues to increase the partners' appearance as a "task group" [14]. Task groups are constituted by partners with shared goals and plans. The puppets' appearance as a task group in the Study 2 contrasted with their appearance as a "loose association" [14] in Study 1, e.g., individuals in close spatial proximity. Five- and 6-year-olds rate task groups as more like "real groups" compared to loose associations [14] and so, e.g., more often expect task group partners to help one another. Study 2 predictions were identical to those of Study 1.

### Methods

**Participants.**   There were 88 participants. The final sample included 64 participants, 32 2-year-olds (*median* = 2.44 years, *range* = 2.25–2.75 years, 11 males) and 32 4-year-olds (*median* = 4.52 years, *range* = 4.25–4.75 years, 21 males). Twenty-four participants had all three trials excluded (*N* = 16 2-year-olds, *N* = 8 4-year-olds). Caregivers predominantly indicated that households made more than $100,000 per year (*N* = 41) or between $60,000-$100,000 per year (*N* = 12). Caregivers predominantly identified participants as White (*N* = 44) or Biracial (*N* = 11). Caregivers received a $10 Amazon gift card. Participants received a certificate. Study design and procedure were approved by the Duke University Campus Institutional Review Board (protocol 2021–0604). Participants were sampled from June 3, 2022 to May 1, 2023. Written informed consent was obtained from participants' parents or legal guardian prior to participation in the procedure described below. Participants were minors and were typically developing (per caregiver statement).

**Design.** The design of the procedure used in Study 2 was identical to that of Study 1.

**Materials.** The materials used in Study 2 were similar to those used in Study 1. However, one large piece of paper was used, instead of several 8 x 11-inch pieces of paper.

**Procedure.** The procedure of Study 2 was similar to that of Study 1. There were three changes made to Study 1. Together, the changes were intended to increase the group entitativity of the puppets (specifically, their appearance as a task group). The first change targeted the participants' introduction to the puppets. In Study 1, the puppets entered individually, without talking, and were greeted in succession. In contrast, in Study 2, the puppets entered together while jointly reminiscing about how much fun they had when they played a previous game together. The second change targeted the puppets' speech at the beginning of the first trial of the Test Phase (i.e., at the start of the key *we* condition). In Study 1, the speaker puppet announced that they wanted to color. The other two puppets were silent. In contrast, in Study 2, the speaker puppet first stated how much fun they had building the blocks (i.e., during the Warmup Phase). This was followed by one comment each from the other two puppets, who also stated how much fun they had building the blocks. Then, the speaker puppet announced that they wanted to color. The third change targeted the materials used in the Test Phase. In Study 1, each puppet was presented with a single sheet of paper in front of them and each puppet colored individually on their sheet of paper. In contrast, in Study 2, a single, large sheet of paper was placed in front of all three puppets and the puppets colored on this single, large sheet of paper.

**Coding.** The coding scheme of Study 2 was identical to that of Study 1. In the final analysis, 135 trials were included (*N* = 64 participants) and 57 trials were excluded (*N* = 38 participants). Of the 135 trials included in the analyses reported, below, 26 participants contributed three trials (78 trials), 19 contributed two trials (38 trials), and 19 contributed one trial (19 trials). All data was initially coded by a research assistant who was blind to the hypotheses of the study. Reliability coding was performed on a random 25% of participants' responses (*N* = 48 trials) by the first author. Responses were coded as in Study 1. Interrater reliability for the number of puppets to whom participants chose to distribute markers was excellent, κ = 0.90, and agreement was excellent, 94% of trials.

**Data analysis plan.** Study 2 data analysis plan was similar to that of Study 1. Like Study 1, a main analysis was followed by follow-up analyses. Like Study 1, the main analysis consisted of a control variables model and a main model. The first three sets of follow-up analyses were identical to those of Study 1. A fourth follow-up analysis analyzed associations between participants' sex and interpretations. Unless otherwise stated, all models in Study 2 included identical predictors and priors as the corresponding Study 1 models.

## Results

**Data.** The distribution of participants' interpretations, by condition, is depicted in Fig 2. Two- and 4-year-olds were split between dyadic and group interpretations. That is, participants tended to distribute markers to the speaker puppet, only, and to the other two puppets at similar rates. Four-year-olds in the *we all* condition made group interpretations nominally more often than any other participant *x* age group pair, with 18 of 26 4-year-olds choosing group interpretations of *we all*. S1 Fig displays the dependent variable as a trichotomous outcome (i.e., according to whether participants chose to distribute markers to 1, 2, or 3 puppets).

**Main analysis.** The posterior control variables model was assessed, first (see S3 Table for posterior parameter estimates). Participants chose group interpretations more often in the third test trial than in the first test trial, 0.49, [-0.19,1.21], $\Pr(\beta > 0.00|D)$ = .92., but not in the second test trial, -0.07, [-0.75,0.60], $\Pr(\beta > 0.00|D)$ = .41. Additionally, participants who heard

*we both* in the second test trial chose group interpretations less often than participants who heard *we all* in their second trial, -0.69, [-1.45,0.04], Pr($\beta > 0.00|D$) = .03. That is, hearing *we both* in the second trial was associated with relatively more dyadic interpretations overall than when *we all* was heard in the second trial. The second exception was the finding that females made group interpretations less often than males, -0.49, [-1.27,0.25], Pr($\beta > 0.00|D$) = .09 (see data in S3 Fig). That is, males made group interpretations more often than females in Study 2. These two findings are handled in subsequent analyses as described below.

The posterior main model was assessed next. Predictions were identical to those of Study 1. The effect of condition order was handled by excluding participants' third trials. This involved removing *N* = 47 trials from the analyses reported, below. By removing the third test trial, the study design became within- and between-subjects: all participants heard *we* in the first trial, but in the second trial participants heard either *we both* or *we all*. Thirty-one participants heard *we both* (*N* = 19 usable trials) in the second test trial and 33 heard *we all* in the second test trial (*N* = 24 usable trials). Only first and second test trial data is reported, below (*N* = 88 trials). See S4 Fig for the posterior parameters of a model fitted to all three test trials' data. The model structure was identical to the main model reported in Study 1.

Based on a model fitted to participants' first and second test trials (Fig 2; data from test trial 1, 2, and 3 in S4 Fig), relatively little support for these predictions was found. There was little indication of an association between age and participants' interpretations of *we*, 0.06, [-0.90,1.05], Pr($\beta$>0.00|$D$) = .54. While 2-year-olds' group interpretations of *we both* and *we all* were nominally less frequent, and nominally more frequent, respectively, than of *we*, neither was a reliable pattern in the posterior model, *we both*: -0.32,[-1.39,0.75], Pr($\beta$>0.00|$D$) = .28; *we all*: 0.43,[-0.67,1.49], Pr($\beta$>0.00|$D$) = .78. There was little evidence that 4-year-olds reliably differed from 2-year-olds, *age * we both*: 0.11,[-1.11,1.29], Pr(β>0.00|$D$) = .57; *age * we all*: 0.16,[-1.02,1.34], Pr(β>0.00|$D$) = .60. In all, these results provide relatively little support for the predictions outlined in the Introduction. The model fitted to the data from all three trials provides similarly little support for the predictions (see S4 Table).

**Follow-up analyses.**   Four sets of follow-up analyses were conducted (first and second trial data, only; marginal posteriors in S5 Table). The first set of follow-up analyses investigated developmental change by condition. There was little evidence for developmental change within conditions, *we*: Pr($\beta$>0.00|$D$) = .44, *we both*: Pr($\beta$>0.00|$D$) = .67, *we all*: Pr($\beta$>0.00|$D$) = .42. This pattern does not accord with the predictions of the present study.

A second set of follow-up analyses investigated condition effects within age groups. There was some evidence that 2-year-olds chose group interpretations less often in response to *we both* than *we*, Pr($\beta$>0.00|$D$) = .23; and more often in response to *we all* than *we*, Pr($\beta$>0.00|$D$) = .76. However, this pattern was not especially robust. In contrast, 4-year-olds chose group interpretations at similar rates in response to *we* and *we both*, Pr($\beta$>0.00|$D$) = .47; but, like 2-year-olds, there was some evidence that 4-year-olds chose group interpretations more often in the *we all* condition than in the *we* condition, Pr($\beta$>0.00|$D$) = .72. Overall, there was some evidence that 2-year-olds interpreted *we*, *we both*, and *we all* differently, more often thinking that *we both* referred to dyads, and *we all* to groups, than *we*. While there was also some evidence that 4-year-olds more often thought that *we all* referred to groups than *we*, they did not distinguish between *we* and *we both*. These results do not support the predictions of Study 2.

A third set of follow-up analyses compared the distribution of participants' responses against chance. All analyses reported in this paragraph excluded third trial data. First, the chance rate of choosing group interpretations was defined as 0.50. Under this definition of chance, 2-year-olds picked group interpretations at chance levels in the *we* condition, Pr($\beta$>0.00|$D$) = .58. However, there was a tendency for 2-year-olds to choose group interpretations at a lower rate after hearing *we both*, Pr($\beta$>0.00|$D$) = .19; and at a higher rate after

hearing *we all*, $\Pr(\beta>0.00|D) = .86$. Four-year-olds, showed a similar, though not identical pattern. Like 2-year-olds, 4-year-olds chose group interpretations at chance levels after hearing *we*, $\Pr(\beta>0.00|D) = .48$; and at higher than chance levels after hearing *we all*, $\Pr(\beta>0.00|D) = .84$. However, in contrast to 2-year-olds, 4-year-olds picked group interpretations at chance levels in response to *we both*, $\Pr(\beta>0.00|D) = .50$. When chance was defined as 0.66, participants typically chose group interpretations below chance levels in response to *we* and *we both*, all $\Pr(\beta_i>0.69|D)<0.11$. However, participants responded markedly closer to chance when interpreting *we all*, 2-year-olds: $\Pr(\beta>0.00|D) = .44$; 4-year-olds: $\Pr(\beta>0.00|D) = .31$. Overall, this third set of follow-up analyses suggested that 2-year-olds distinguish between *we*, *we both*, and *we all* by favoring dyadic interpretations of *we both*, group interpretations of *we all*, and dyadic or group interpretations *we*. Interestingly, 4-year-olds treated *we* like *we both* (by choosing dyadic or group interpretations) but favored group interpretations of *we all*. None of the directional associations were particularly robust in the posterior models.

A fourth set of follow-up analyses investigated the association of interpretations with participant sex. Two models were fitted. In both, participant sex was the only fixed effects predictor. In the first two trials, females chose group interpretations less often than males (Fig 3), $\Pr(\beta>0.00|D) = .30$. However, this pattern was not especially robust. This pattern became more robust when all three trials were examined, $\Pr(\beta>0.00|D) = .12$. A series of three further models, structurally identical to those above, were fitted in each condition (first two test trials, only). The strongest posterior association was in the *we* condition, $\Pr(\beta>0.00|D) = .28$; *we both*: $\Pr(\beta>0.00|D) = .53$; *we all*: $\Pr(\beta>0.00|D) = .43$. That is, while it was not particularly robust, the strongest evidence for an association between interpretation and participant sex was for females to choose group interpretations less often than males in the *we* condition. Overall, these results may suggest that males more readily form group interpretations than females, at least when interpreting the linguistic forms used in the sort of referring situation employed in this study.

In sum, the results of the follow-up analyses of Study 2 suggest little evidence for developmental change in interpretation of *we*, *we all*, or *we both*. The strongest evidence that participants distinguished conditions was found when examining their interpretations against chance. In this case, there was some evidence that 2- and 4-year-olds interpreted *we* as referring to dyads half the time and groups half the time. However, while 4-year-olds interpreted *we both* similarly as they did *we*, 2-year-olds tended to choose dyadic interpretations of *we both*. Participants generally interpreted *we all* as referring to groups. Males may make group interpretations more often than females, but this cannot be determined conclusively at present.

## Study 2 Discussion

Study 2 investigated children's interpretations of *we*, *we both*, and *we all* in a procedure modified to increase their tendency to choose group interpretations, relative to Study 1. As expected, participants were nominally more likely to choose group interpretations in Study 2 compared to Study 1. Moreover, in Study 2, there was some evidence that suggested that 2-year-olds favor dyadic interpretations of *we both*, and group interpretations of *we all*, relative to *we*. This pattern was counter to 2-year-olds' behavior in Study 1. For their part, 4-year-olds in Study 2 did not distinguish in their choices between *we*, *we both*, and *we all*. Thus, the effect of condition on 4-year-olds' interpretations was similar in Study 1 and Study 2. Participants' behavior in Study 2 was counter to predictions– 2-year-olds did not treat *we* like *we both* and not *we all*, and 4-year-olds did not treat *we* like *we all* and not *we both*. There was some

evidence that males chose group interpretations more often than females, perhaps especially in the *we* condition.

The following section presents an analysis that compared participants' behavior in Study 1 and Study 2. Note that the procedures of the two studies were essentially identical (save for three minor tweaks to the procedure of Study 2 that were intended to increase the group entitativity of participants' puppet partners; see Methods, Study 2). Thus, it was deemed justified and informative to compare behavior in the two studies within one model. This analysis informs about effects of group entitativity on interpretation of *we*, *we both*, and *we all*.

## Comparison of Study 1 and Study 2

### Methods

**Dataset combined from Study 1 and Study 2.**  There were 128 participants (*N* = 64 from Study 1 and Study 2, respectively). These participants participated in either Study 1 or Study 2. There were 64 2-year-olds (*median* = 2.48 years, *range* = 2.25–2.75 years, 29 males) and 64 4-year-olds (*median* = 4.53 years, *range* = 4.25–4.75 years, 34 males). There were 288 trials included in the analyses reported, below (*N* = 153 from Study 1 and *N* = 135 from Study 2). Of the 288 included trials, 64 participants contributed three trials (192 trials), 32 contributed two trials (64 trials), and 32 contributed one trial (32 trials).

**Data analysis plan.**  The data analysis plan was similar to those of the studies, above. A main analysis included a control variables model and main model. There was one set of follow-up analyses.

*Main analysis.* A control variables model, identical to those presented above, was fitted to the data from both studies. The main model fitted to the combined dataset expanded the main models presented above. Three facts justified model expansion. First, there was a substantially larger dataset used in this analysis, relative to the analyses presented in Study 1 and Study 2. This enabled more accurate inference of higher-order interactions. Second, there were suggestive relations between participant sex and interpretation reported in the analysis of Study 2. This called for the analysis of higher-order interactions between sex and other predictors. Third, there was the key requirement of analyzing the association between participants' interpretations and study (*qua* "manipulation" of puppets' apparent group entitativity). This called for the inclusion of study as a predictor term in the model. The main model specification was as follows:

$$\text{interpretation} \sim \text{sex} * \text{age group} * \text{condition} * \text{study} + (\text{condition}||\text{participant})$$

The model included main effects of sex, age group, condition, and study. Model expansion fell upon the inclusion of all interactions between sex, age, condition, and study, to fourth order. Random effects and priors were identical to Study 1 and Study 2 main model. Predictors were treatment coded. Posterior parameters were characterized as in Study 1 and Study 2.

*Follow-up analyses.* One set of follow-up analyses investigated the effect of study on participants' interpretations. Three identical models were fitted to participants' data, one for each condition. Model structure included main effects of study and age and their interaction.

### Results

**Main analysis.**  The posterior control variables model was assessed, first (see S6 Table for posterior parameter estimates). As in Study 2, there was some evidence for an order effect, -0.46,[-1.18,0.22], Pr($\beta$>0.00|*D*) = .09. Participants who heard *we both* in the second test trial chose group interpretations less often than participants who heard *we all* in their second trial.

The posterior main model was assessed next. Participants were predicted to favor group interpretations of *we* in Study 2 compared to Study 1. The effect of order of conditions was handled by excluding the third test trial. This involved removing $N = 101$ trials from the analysis, below. Sixty-five participants heard *we both* ($N = 47$ usable trials) in the second test trial and 63 heard *we all* in the second test trial ($N = 46$ usable trials). Only first and second test trial data is reported in the analyses, below ($N = 187$ trials). See S7 Table for the posterior parameters of a model fitted to all three test trials' data.

Participants in Study 2 chose group interpretations reliably more often than participants in Study 1, 0.90,[-0.15,1.96], Pr($\beta$>0.00|$D$) = .95. That is, the changes enacted to the procedure in Study 2 –intended to increase the group entitativity of participants' partners–increased participants' tendency to choose group interpretations compared to Study 1. There were only weak posterior associations among all other predictors (see S8 Table for marginal posteriors). The model fitted to all three test trials showed an identical pattern of posterior associations (see S7 Table). This pattern provided mixed support for our predictions. The group entitativity of partners increased participants' tendency to assign group interpretations overall, rather than only to *we*.

**Follow-up analyses.** One set of follow-up analyses investigated the effect of group entitativity on participants' interpretations. Following the reliable posterior order effect, above, data was analyzed from test trials 1 and 2, only (see S9 Table for marginal posteriors). Participants in Study 2 more often chose group interpretations of *we* than participants in Study 1, 0.94, [0.04,1.85], Pr($\beta$>0.00|$D$) = .98. This association did not reliably depend on participant age, 0.05,[-0.95,1.07], Pr($\beta$>0.00|$D$) = .54. That is, both 2- and 4-year-olds chose group interpretations of *we* more often in Study 2 than in Study 1. Participants' interpretations of *we both* and, to a lesser degree, *we all* did not reliably depend on study, *we both*: 0.25,[-0.77,1.28], Pr($\beta$>0.00|$D$) = .68; *we all*: 0.58,[-0.44,1.60], Pr($\beta$>0.00|$D$) = .87. Neither of these latter associations depended reliably on participant age, *we both*: 0.35, [-0.84,1.53], Pr($\beta > 0.00$|$D$) = .72; *we all*: 0.16, [-0.98,1.29], Pr($\beta > 0.00$|$D$) = .61. Overall, these results accorded with predictions. Group entitativity only reliably impacted participants' interpretations of *we*, with greater group entitativity pulling for relatively more group interpretations. Participants' interpretations of *we both* and *we all* were not reliably impacted by group entitativity, given the model and data.

## General discussion

The present studies investigated whether a groupminded shift in 3-year-olds' social conceptualization [7] influences linguistic interpretation. To this end, a novel, virtual paradigm was leveraged to investigate children's interpretation of the pragmatically ambiguous English pronoun *we*. It was predicted that 2-year-olds would favor dyadic interpretations of *we*–that is, that their interpretations of *we* and *we both* would be similar–and that 4-year-olds would favor group interpretations of *we*–that is, that their interpretations of *we* and *we all* would be similar. It was further predicted that increasing the group entitativity [14] of children's partners would increase children's tendency to choose group interpretations of *we*, but not *we both* or *we all*. Two studies, and a comparison of their results, were conducted to investigate these predictions. In our view, the most interesting result of the present studies was found in the effect of group entitativity on children's interpretation of *we*. This finding is discussed, below, after discussion of the results of Study 1 and Study 2, respectively.

In Study 1, participants generally chose dyadic interpretations of *we*, *we both*, and *we all*. This pattern was counter to predictions. Indeed, surprisingly, the only group that chose dyadic interpretations at less than chance rates was 2-year-olds in the *we all* condition. While it was

expected that both age groups would choose group interpretations at elevated rates in response to the unambiguous *we all*, it is curious that only 2-year-olds, and not 4-year-olds, behaved in this way. It could be argued that participants' tendency to choose dyadic interpretations in response to *we both* aligned with our predictions. However, for us, the fact that participants also favored dyadic interpretations of *we* undermines this interpretation. Rather, it was suspected that the contextual cues present in the procedure of Study 1 pulled too strongly for dyadic interpretations. Consequently, 2- and 4-year-olds generally favored dyadic interpretations; though it is unclear why 2-year-olds chose group interpretations of *we all* at elevated rates. Nonetheless, this "contextual cues interpretation" motivated the design of Study 2.

In Study 2, participants were nominally more likely to choose group interpretations of *we*, *we both*, and *we all* than participants in Study 1. On its face, this suggests that the changes enacted to the procedure in Study 2 worked as intended by increasing the group entitativity of participants' partners. However, as concerns the anticipated condition effects, matters were complicated by the presence of an order effect: Participants who heard *we both* before *we all* were more likely to choose dyadic interpretations of *we both* and *we all* than were participants who heard *we all* before *we both*. This was not the case in Study 1 and, speculatively, may be related to the increased group entitativity of participants' partners. After excluding the third trial, analyses of participants' behavior in the Study 2 suggested that, as in Study 1, there was little in the way of statistically reliable behavioral distinctions made by participants between conditions. Again, this pattern was against predictions.

Interestingly, in Study 2, there was some evidence for an association between participant sex and interpretations. Males chose group interpretations more often than females, although the evidence for this was not conclusive. Nonetheless, this pattern aligns with prior research. First, studies of pronoun development suggest that males may form group interpretations of *we* more often than females [11]. Second, research on young children's collaboration suggests that, compared to females, males are more parochial in their sharing [24–26] and more committed towards collaborators [23, 27]. Nonetheless, three caveats suggest that additional research is needed before issuing a conclusive statement about the relationship of sex and children's interpretation of the forms investigated in this study. First, the relationship with sex was strongest in an analysis of all three trials and, as noted above, this was confounded with an order effect. Second, neither Study 2 nor Study 1 were intended to investigate sex-based interpretational differences. Third, the relationship between sex and interpretation, whatever its qualities, was present only in Study 2 and not in Study 1.

Of greatest interest, a comparison of the present studies found a robust association between interpretations and the group entitativity of participants' partners. There are multiple substrates supportive of group entitativity, that is, the appearance of a collection of individuals as a singular "we"–individuals may share interests, form a social category, be spatially aggregated, or, as in Study 2, form a collaborative "task group" [14]. In a comparison of the present studies, participants chose group interpretations more often when their partners appeared to be a group (Study 2) instead of a collection of individuals (Study 1). While the relevant interaction terms in the main model were not reliably signed, a follow-up analysis suggested that increased group entitativity caused participants to reliably choose group interpretations of *we* more often, only, and not *we both* or *we all*. This pattern aligned with predictions. One interpretation of this pattern is that participants who saw the puppets as a group entity (in Study 2) relied more heavily on nonlinguistic contextual cues to resolve pragmatic ambiguity about the intended referent of *we* compared to *we both* or *we all* [28]. Importantly, however, one limitation of the present study limits how far we wish to push any specific interpretation of this result, at present.

Perhaps the main limitation of the present research was the materials. Future research might consider using materials different than those used in the present studies. These materials required children to know how to interpret uses of the phrases *we both* and *we all*, or at least of the quantifiers contained in those phrases. However, the frequency of these forms in young children's input is unknown. Thus, it is possible that our failure to find the predicted results owed, at least in part, to participants' having not known the phrases *we both* and *we all*, or at least their constituent quantifiers. In addition, one might view these terms as inviting children to draw scalar implicatures about speaker-intended meaning, something with which young children struggle [29]. Moreover, young children struggle to appropriately understand uses of universal quantifiers, like *all*, even in situations in which scalar implicatures do not have to be drawn [30]. Relatedly, it is plausible that the departure of children's interpretation of *we both* and *we all* from predictions does not have to do with their inexperience with these forms. Instead (or in addition), children's conceptual skills may be involved. Checking this requires additional control conditions that enable comparison of children's interpretation of *we* and its quantified forms to that of forms like *you and I* or *you, me, and Lion*. If children's interpretations of those forms aligns with their interpretations of *we both* and *we all*, as in the present studies, then this may suggest that children's conceptual skills are implicated in explaining the results of the present studies.

Taken together, the present studies demonstrate that young children's interpretation of bare *we* is impacted by the apparent entitativity of the group of which the speaker is part. This much is clear. However, it is unclear why this is the case. Is it because young children rely more on relevant nonlinguistic cues to resolve referential ambiguity for *we* compared to *we both* or *we all*, or because they simply do not know these latter forms?

Two related points are offered as alternative, or perhaps supplementary, explanations for the present results. It is important to consider, first, children's experience hearing how others use *we* and, second, the possibility that–even after they have gained their groupminded conceptual skills–it may take children some time before they realize that group interpretations may be assigned to uses of *we*. Regarding the first point, it is plausible that children typically hear *we* used with dyadic referential intentions (e.g., when a caregiver informs their child what "we" are doing today) and gain increased exposure to group referential intentions as they age (e.g., when a teacher informs their pupils what "we" are doing today). If this is the case, it may impact the interpretations children assign to uses of *we* and, consequently, their behavior in the present research. As for the second point, children plausibly take some unknown amount of time to realize–after they have gained their groupminded conceptual skills–that group referential intentions underlie others' uses of *we* in certain instances. This process might be thought of as coming to some kind of realization–enabled by their newfound groupminded conceptual skills–that a new and distinct class of (group) referents may be designated by others' uses of *we*. Moreover, these two points may be related to one another at least insofar as children must, first, realize that *we* may be used with group referential intentions before they can capitalize on potentially increased exposure to group-referring uses of *we*. In any case, these points, and their putative relationship, are speculative because there is no directly relevant research. Thus, the uses of *we* to which children are exposed, whether there is a learning process enabled by maturationally constrained developments in conceptual structure, and whether these are related in ontogeny, are questions for future research.

Future research might also investigate why most of the major predictions of the present studies were not supported. There is substantial research that supports the notion of a group-minded shift at 3 years of age [7], including research on children's pronoun production [12] and interpretation of exclusive *we* [11]. It is therefore not entirely clear, at present, why the present studies failed to find support for the predicted pattern of interpretation of inclusive *we*.

It is possible that the predicted pattern for children is, in fact, not the actual one. A check on this is to conduct an in person analogue of the present studies. This is important because the pragmatics of the current procedure was stilted. Children had to infer that they were part of an intended, first-person plural referent that included individuals who were together, onscreen. Children had to infer that they were part of a group with individuals displayed onscreen. This situation is not representative of most children's naturalistic interactions, which are primarily in person and not online. It will be important, too, to include adults as a control group. This is necessary to ensure that the expected adult interpretations of *we both* and *we all* do, in fact, obtain. Additional research might investigate the relation between children's sex and their interpretation of *we*–do males, for instance, form group interpretations more readily than females? If so, what is the ontogenetic trajectory? Future research might also expand upon the present group entitativity results. Does the same pattern obtain when group entitativity is manipulated within one study? Does the same pattern obtain when group entitativity is instantiated differently, e.g., as an interest group or a social category? Finally, future research might investigate potential relations between children's interpretation of *we* and their understanding of the symbolic representation of numerosity. Research indicates that, at around 3 years of age, powerful skills emerge that enable children to link linguistic forms that symbolize cardinality with the number of elements in a set [31], and this has consequences for their ability to count to numbers larger than three.

In conclusion, the present article reported the results of two studies aimed at understanding the relation between age-related shifts in children's social-conceptual skills and their pronoun interpretation. While evidence was lacking for predicted relations between children's age and their interpretations of *we*, *we both*, and *we all*, children interpreted these forms as referring to a group more often when speakers appeared to be part of a group than when they did not. To our knowledge, this is the first evidence of a relationship between group entitativity and children's interpretation of pronominal forms.

## Supporting information

**S1 Fig. Trial-level distribution of trichotomous dependent variable, by condition.**
(TIF)

**S2 Fig. Participant-level distribution of dichotomous dependent variable, by participant sex.**
(TIF)

**S3 Fig. Participant interpretations, by sex, in test trials 1, 2, and 3, Study 2.**
(TIF)

**S4 Fig. Participant interpretations in test trials 1, 2, and 3, Study 2.**
(TIF)

**S1 Table. Posterior parameter estimates of control variables model, Study 1.**
(DOCX)

**S2 Table. Marginal posterior distributions of models reported in the three follow-up analyses of Study 1.** "Estimate" represents median, "Error" represents 1 SD, "HDI" represents the 95% highest density interval.
(DOCX)

**S3 Table. Posterior parameter estimates of control variables model, Study 2.**
(DOCX)

**S4 Table. Posterior parameters of a model fitted to participants' test trial 1, 2, and 3 data, Study 2.**
(DOCX)

**S5 Table. Marginal posterior distributions of parameters in models reported in the four follow-up analyses of Study 2.** "Estimate" represents median, "Error" represents 1 SD, "HDI" represents the 95% highest density interval.
(DOCX)

**S6 Table. Posterior parameter estimates of control variables model, comparison of Study 1 and Study 2.**
(DOCX)

**S7 Table. Posterior parameters of a model fitted to participants' test trial 1, 2, and 3 data, comparison of Study 1 and Study 2.**
(DOCX)

**S8 Table. Posterior parameters of a model fitted to participants' test trial 1 and 2 data, comparison of Study 1 and Study 2.**
(DOCX)

**S9 Table. Marginal posterior distributions of parameters in models reported in the follow-up analysis of the comparison of Study 1 and Study 2.**
(DOCX)

## Acknowledgments

We thank Alexandra Springer and Isabelle Ginn for their role in data collection in the studies reported in the present article. We thank Alissa Rivero for her role in coding children's behavior in the studies reported in the present article.

## Author Contributions

**Conceptualization:** Jared Vasil, Camryn Capoot, Michael Tomasello.

**Data curation:** Jared Vasil.

**Formal analysis:** Jared Vasil.

**Funding acquisition:** Michael Tomasello.

**Investigation:** Camryn Capoot.

**Methodology:** Camryn Capoot.

**Project administration:** Jared Vasil, Camryn Capoot.

**Software:** Jared Vasil.

**Supervision:** Jared Vasil, Michael Tomasello.

**Visualization:** Jared Vasil.

**Writing – original draft:** Jared Vasil.

**Writing – review & editing:** Jared Vasil, Michael Tomasello.

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
