## [Decision Letter · Decision Letter 0]

17 Oct 2023

PONE-D-23-24720Young English-speaking Children’s Interpretation of Inclusive We: Effects of Group EntitativityPLOS ONE

Dear Dr. Vasil,

Thank you for submitting your manuscript to PLOS ONE. After careful consideration, we feel that it has merit but does not fully meet PLOS ONE’s publication criteria as it currently stands. Therefore, we invite you to submit a revised version of the manuscript that addresses the points raised during the review process.

We look forward to receiving your revised manuscript.

Kind regards,

Barbara T Rumain, PhD

Academic Editor

PLOS ONE

Additional Editor Comments:

Reviewer 1:

This paper reports the results of two virtual experiments examining the interpretation of the inclusive pronoun “we” by 2- and 4-year-olds. The two experiments differed in the context of the task. In the first experiment, the partners (and potential referents of the pronoun “we”) appeared like an aggregation of individuals, whereas in the second, they were presented as a collaborative group. The results showed that children more often interpret “we” as referring to a group in a collaborative group context (second experiment) than when it was not the case (first experiment) but, and it is a weakness of the paper, it is the only reliable result of the study. However, the paper is clear and well-written, the study is undertaken in a careful manner and the data are handled appropriately.

Minor comments:

- Naming the two experiments “pilot experiment” and “main experiment” respectively is not really justified, so why not just name them Experiment 1 and Experiment 2.

- It is rather surprising that the authors refer to the variable sex when they are apparently referring to the gender of the participants. Unless they are referring solely to biological and physiological differences between males and females, the variables sex and gender are at least confounded here. This point needs to be clarified.

- The methodology sections are a little confusing to read. In the first experiment, in the participants section, it is mentioned that seven participants were excluded and then, in the coding section, it is mentioned that 60 trials were excluded, which seems to take into account the trials of the participants excluded above. Why not mention here only the trials excluded from the trials of the 64 participants retained for the study? Same comment for the second experiment.

- Still regarding excluded trials, “Of the 60 excluded 9 trials, 7 participants contributed 3 trials (21 trials), 13 contributed 2 trials (26 trials), and 13contributed 1 trial (13 trials)” (p.11). Is this information really important? And if so, what does it contribute? If not, I suggest dropping this information. Same comment for the second experiment.

- Please do not use E for experimenter or put it in brackets when first mentioned. Idem for P for participant.

- I'm not familiar with Bayesian models (and I guess I'm not the only one), how were the priors chosen (p.12)?

- As underlined by the authors in the general discussion (p.32), the variable sex/gender was not purposely manipulated (in both experiments, there were far fewer males than females) and the observed effects of sex/gender were statistically weak. The effects of this variable are therefore not worth mentioning in the abstract and in the final conclusion (p.35).

Reviewer 2:

The data doesn't support all of the conclusions drawn. The results are potentially confounded in a few different ways (detailed in my review) and yet the authors - while very open and and honest about what the stats show - still conclude mostly with the hypothesis and don't consider reasonable alternatives that may explain their results (which were mostly unexpected relative to the hypothesis)

**NOTE: REVIEWER 2 ALSO SUBMITTED A 4-PAGE pdf WITH COMMENTS. PLEASE SEE THIS pdf, WHICH IS ATTACHED.**

Reviewers' comments:

Reviewer's Responses to Questions

**Comments to the Author**

1. Is the manuscript technically sound, and do the data support the conclusions?

Reviewer #1: Yes

Reviewer #2: Partly

2. Has the statistical analysis been performed appropriately and rigorously? 

Reviewer #1: Yes

Reviewer #2: Yes

3. Have the authors made all data underlying the findings in their manuscript fully available?

Reviewer #1: Yes

Reviewer #2: Yes

4. Is the manuscript presented in an intelligible fashion and written in standard English?

Reviewer #1: Yes

Reviewer #2: Yes

5. Review Comments to the Author

Reviewer #1: This paper reports the results of two virtual experiments examining the interpretation of the inclusive pronoun “we” by 2- and 4-year-olds. The two experiments differed in the context of the task. In the first experiment, the partners (and potential referents of the pronoun “we”) appeared like an aggregation of individuals, whereas in the second, they were presented as a collaborative group. The results showed that children more often interpret “we” as referring to a group in a collaborative group context (second experiment) than when it was not the case (first experiment) but, and it is a weakness of the paper, it is the only reliable result of the study. However, the paper is clear and well-written, the study is undertaken in a careful manner and the data are handled appropriately.

Minor comments:

- Naming the two experiments “pilot experiment” and “main experiment” respectively is not really justified, so why not just name them Experiment 1 and Experiment 2.

- It is rather surprising that the authors refer to the variable sex when they are apparently referring to the gender of the participants. Unless they are referring solely to biological and physiological differences between males and females, the variables sex and gender are at least confounded here. This point needs to be clarified.

- The methodology sections are a little confusing to read. In the first experiment, in the participants section, it is mentioned that seven participants were excluded and then, in the coding section, it is mentioned that 60 trials were excluded, which seems to take into account the trials of the participants excluded above. Why not mention here only the trials excluded from the trials of the 64 participants retained for the study? Same comment for the second experiment.

- Still regarding excluded trials, “Of the 60 excluded 9 trials, 7 participants contributed 3 trials (21 trials), 13 contributed 2 trials (26 trials), and 13contributed 1 trial (13 trials)” (p.11). Is this information really important? And if so, what does it contribute? If not, I suggest dropping this information. Same comment for the second experiment.

- Please do not use E for experimenter or put it in brackets when first mentioned. Idem for P for participant.

- I'm not familiar with Bayesian models (and I guess I'm not the only one), how were the priors chosen (p.12)?

- As underlined by the authors in the general discussion (p.32), the variable sex/gender was not purposely manipulated (in both experiments, there were far fewer males than females) and the observed effects of sex/gender were statistically weak. The effects of this variable are therefore not worth mentioning in the abstract and in the final conclusion (p.35).

Reviewer #2: The data doesn't support all of the conclusions drawn. The results are potentially confounded in a few different ways (detailed in my review) and yet the authors - while very open and and honest about what the stats show - still conclude mostly with the hypothesis and don't consider reasonable alternatives that may explain their results (which were mostly unexpected relative to the hypothesis)

6. PLOS authors have the option to publish the peer review history of their article (what does this mean?). If published, this will include your full peer review and any attached files.

Reviewer #1: No

Reviewer #2: No

---

## [Author Response · Author response to Decision Letter 0]

7 Dec 2023

Reviewer 1

This paper reports the results of two virtual experiments examining the interpretation of the inclusive pronoun “we” by 2- and 4-year-olds. The two experiments differed in the context of the task. In the first experiment, the partners (and potential referents of the pronoun “we”) appeared like an aggregation of individuals, whereas in the second, they were presented as a collaborative group. The results showed that children more often interpret “we” as referring to a group in a collaborative group context (second experiment) than when it was not the case (first experiment) but, and it is a weakness of the paper, it is the only reliable result of the study. However, the paper is clear and well-written, the study is undertaken in a careful manner and the data are handled appropriately.

Minor comments:

Naming the two experiments “pilot experiment” and “main experiment” respectively is not really justified, so why not just name them Experiment 1 and Experiment 2.

We thank the Reviewer for their attention to our manuscript and for their thoughtful commentary and feedback on the original draft of the manuscript.

All uses of Pilot Study and Main Study have been replaced with Study 1 and Study 2, in the Main Text, Supplementary Material, and all tables and figures.

It is rather surprising that the authors refer to the variable sex when they are apparently referring to the gender of the participants. Unless they are referring solely to biological and physiological differences between males and females, the variables sex and gender are at least confounded here. This point needs to be clarified.

We were referring to the sex of the participants because we asked participants’ caregivers only what sex their child is. Gender was never inquired so we do not refer to it.

The methodology sections are a little confusing to read. In the first experiment, in the participants section, it is mentioned that seven participants were excluded and then, in the coding section, it is mentioned that 60 trials were excluded, which seems to take into account the trials of the participants excluded above. Why not mention here only the trials excluded from the trials of the 64 participants retained for the study? Same comment for the second experiment.

After rereading this portion of the manuscript, we agreed with the Reviewer’s suggestion. We have rephrased the manuscript in line with the Reviewer’s suggestion (revisions on pp. 11 and 22).

From the text of the first study, the original text read: “In the final analysis, 153 trials were included (N = 64 participants) and 60 trials were excluded (N = 33 participants). Of the 153 included trials, 38 participants contributed 3 trials (114 trials), 13 contributed 2 trials (26 trials), and 13 contributed 1 trial (13 trials). Of the 60 excluded trials, 7 participants contributed 3 trials (21 trials), 13 contributed 2 trials (26 trials), and 13 contributed 1 trial (13 trials).”

The revised text (p. 11) now states: “In the final analysis, 153 trials were included (N = 64 participants) and 39 trials were excluded (N = 26 of the 64 participants). Of the 153 included trials, 38 participants contributed 3 trials (114 trials), 13 contributed 2 trials (26 trials), and 13 contributed 1 trial (13 trials).”

From the text of the second study, the original text read: “In the final analysis, 135 trials were included (N = 64 participants) and 129 trials were excluded (N = 62 participants). Of the 135 trials included in the analyses reported, below, 26 participants contributed 3 trials (78 trials), 19 contributed 2 trials (38 trials), and 19 contributed 1 trial (19 trials). Of the 129 excluded trials, 24 participants contributed 3 trials (72 trials), 19 contributed 2 trials (38 trials), and 19 contributed 1 trial (19 trials).”

The revised text (p. 22) now states: “In the final analysis, 135 trials were included (N = 64 participants) and 57 trials were excluded (N = 38 participants). Of the 135 trials included in the analyses reported, below, 26 participants contributed 3 trials (78 trials), 19 contributed 2 trials (38 trials), and 19 contributed 1 trial (19 trials).”

Still regarding excluded trials, “Of the 60 excluded 9 trials, 7 participants contributed 3 trials (21 trials), 13 contributed 2 trials (26 trials), and 13contributed 1 trial (13 trials)” (p.11). Is this information really important? And if so, what does it contribute? If not, I suggest dropping this information. Same comment for the second experiment.

After rereading these portions of the manuscript, we agree with the Reviewer’s suggestion. We have removed these lines in the revised manuscript (please see previous response).

Please do not use E for experimenter or put it in brackets when first mentioned. Idem for P for participant.

The first use of E for experimenter was placed in brackets (p. 7). Additionally, P for participant has been replaced with “participant” in Figure 1 (p. 10); this was the only location in the manuscript in which this form was used.

I'm not familiar with Bayesian models (and I guess I'm not the only one), how were the priors chosen (p.12)?

We respond in two parts. First, we note that we have revised the manuscript text by adding the following text: “Priors were chosen based on general domain knowledge and an understanding of the response distribution defined by the link function so as to be “weakly informative” about the most likely outcomes. This strategy is considered best practice in Bayesian modeling (Gelman et al., 2013; Lemoine, 2019; Schad et al., 2021) and has been used in previous developmental psychological analyses (Vasil et al., forthcoming; Vasil & Tomasello, 2022),” (p. 13).

Second, we elaborate on the revised text. As noted, priors were chosen based on domain knowledge and an understanding of the response distribution defined by the link function so as to be “weakly informative” about the posterior distribution. This is considered best practice in Bayesian modeling and has been used in previous analyses of development psychological experiments. For example, general domain knowledge (e.g., from previous child development studies) suggests that, though not impossible, it is unlikely that a manipulation will switch binary choice behavior from 99% response X to 99% response Y. This intuition can be formalized as a (prior) probability distribution over the parameter space of the response distribution with maximum likelihood on 0 (i.e., that signals no effect, chance, etc.). When the logit function is used to link the predictor and response distributions, values greater than 1.5 or less than -1.5 (i.e., plus/minus two standard deviations of the SD of the prior distribution) are considered a priori highly unlikely, though not impossible. That is, predictor odds ratio values that have an absolute value greater than 1.5 lie outside the set of 95% a priori most likely odds ratios. Heuristically, the logic behind Bayesian methods can be understood as generalizing frequentist methods from inference about point estimates to inference about the distribution of point estimates (Gelman et al., 2020).

The same logic used to define the prior distributions of the predictor variables was applied to definitions of the prior distributions over the other parameters, including the intercept and hierarchical variance parameters.

As underlined by the authors in the general discussion (p.32), the variable sex/gender was not purposely manipulated (in both experiments, there were far fewer males than females) and the observed effects of sex/gender were statistically weak. The effects of this variable are therefore not worth mentioning in the abstract and in the final conclusion (p.35).

We agree with the Reviewer that associations with sex should not be mentioned in the abstract and final conclusion. Accordingly, we have removed mention of sex in the abstract and in the final conclusion (p. 36). We have additionally removed what we considered to be redundant text (p. 20). That text originally read: “There was little evidence for associations between other predictors and participant interpretations; for example, both sexes favored dyadic interpretations.” The revised text excludes the portion of text after the semicolon.

Moreover, we have fixed an error in reporting of sex demographics of the comparison study. Originally, it was stated that 11 2-year-olds males and 21 4-year-old males were included in the Comparison of Study 1 and Study 2 analyses. These numbers were erroneously imported from (correct) reporting of those same numbers in the participants section of Study 2 (p. 21). The revised manuscript corrects this error by reporting the added values from studies 1 and 2. Thus, there are reported to be 29 2-year-old males and 34 4-year-old males in the comparison analyses (p. 28).

References

Gelman, A., Carlin, J., Stern, H., Dunson, D., Vehtari, A., & Rubin, D. (2013). Bayesian Data Analysis (3rd ed.). CRC Press/Taylor & Francis.

Gelman, A., Hill, J., & Vehtari, A. (2020). Regression and Other Stories. Cambridge University Press.

Lemoine, N. P. (2019). Moving beyond noninformative priors: Why and how to choose weakly informative priors in Bayesian analyses. Oikos, 128(7), 912–928. https://doi.org/10.1111/oik.05985

Schad, D. J., Betancourt, M., & Vasishth, S. (2021). Toward a principled Bayesian workflow in cognitive science. Psychological Methods, 26(1), 103–126. https://doi.org/10.1037/met0000275

Vasil, J., Price, D., & Tomasello, M. (forthcoming). Thought and Language: Effects of Groupmindedness on Young Children’s Interpretation of We. Child Development.

Vasil, J., & Tomasello, M. (2022). Effects of “we”-framing on young children’s commitment, sharing, and helping. Journal of Experimental Child Psychology, 214, 105278. https://doi.org/10.1016/j.jecp.2021.105278

 

Reviewer 2

The data doesn't support all of the conclusions drawn. The results are potentially confounded in a few different ways (detailed in my review) and yet the authors - while very open and honest about what the stats show - still conclude mostly with the hypothesis and don't consider reasonable alternatives that may explain their results (which were mostly unexpected relative to the hypothesis)

We thank the Reviewer for their attention to our manuscript and for their thoughtful commentary and feedback on the original draft.

As the only positive conclusion we drew from the study was the relationship between group entitativity (Study 1 vs. Study 2) and interpretation of we, we both, and we all, we understood the Reviewer’s concern to pertain to this conclusion. In the original manuscript, we concluded that children’s interpretation of we, but of neither we both nor we all, was influenced by group entitativity (with “entitative” groups pulling more for group rather than dyadic interpretations). Moreover, we presented a subsequent paragraph that elaborated an interpretation of this conclusion.

After carefully rereading and reconsidering these portions of text, we agree that our conclusion was not fully supported by the evidence. In light of other Reviewer comments (below), we have revised this portion of text. The subsequent interpretation paragraph, noted above, was removed and replaced with a single sentence that more tentatively suggests the same interpretation. Moreover, this is followed by a new paragraph that relates our revised, more careful conclusion to limitations in the study design noted by the Reviewer. Specifically, in the revised text, we now state (p. 34):

“One interpretation of this pattern is that participants who saw the puppets as a group entity (in Study 2) relied more heavily on nonlinguistic contextual cues to resolve pragmatic ambiguity about the intended referent of we compared to we both or we all (Vasil, 2023). Importantly, however, one limitation of the present study limits how far we wish to push any specific interpretation of this result, at present. 

Perhaps the main limitation of the present research was the materials used in the present study. Future research might consider using materials different than those used in this research. The materials used in the present studies required children to know how to interpret uses of the phrases we both and we all, or at least of the quantifiers contained in those phrases. However, the frequency of these forms in young children’s input is unknown. Thus, it is possible that our failure to find the predicted results owed, at least in part, to participants’ having not known the phrases we both and we all, or at least their constituent quantifiers. In addition, one might view these terms as inviting children to draw scalar implicatures about speaker-intended meaning, something with which young children struggle (Papafragou & Skordos, 2016). Moreover, young children struggle to appropriately understand uses of universal quantifiers, like all, even in situations in which scalar implicatures do not have to be drawn (Brooks & Braine, 1996). Taken together, the present studies demonstrate that young children’s interpretation of bare we is impacted by the apparent entitativity of the group of which the speaker is part. This much is clear. However, it is unclear why this is the case. Is it because young children rely more on relevant nonlinguistic cues to resolve referential ambiguity for we compared to we both or we all, or because they simply do not know these forms?” 

Lack of developmental linguistic consideration of quantifiers in stimuli (“both” and “all”)

The Reviewer suggests several insightful limitations of the present research. After carefully rereading the original manuscript, we have decided to introduce discussion of some of these limitations in the revised manuscript (reproduced above from p. 34 of the revised manuscript). We add only that it is possible that children, or some children, learn the use of such quantifiers as all and both in the forms we all and we both not as analyzed, constituent parts of quantified first-person plurals (e.g., [we] [both]) but, instead, as unanalyzed, frozen phrases (e.g., [we both]). We have alerted readers to both possibilities in the revised text.

Lack of good control condition, and control population.

We first respond to the Reviewer’s comment about a no-we¬ control condition before responding to their comment about an adult control population. The Reviewer’s point about a comparison between a hypothetical no-we¬ condition and the we condition is well taken. We agree that it would be an interesting comparison. However, while our focus in the manuscript was on the relationship between group-mindedness and language development, we do not claim that such effects of group-mindedness are unique to language. In fact, we believe that these effects are not unique to language, and evidence for this is reviewed in Introduction. However, nothing in our research question hangs on how group-mindedness influences other facets of psychological development. Therefore, we consider a no-we control condition to be outside the scope of the study. We suggest that, if we changed the research question, or retained the same research question but claimed that the predicted effects were unique to language, then, in that case, we agree that a no-we control condition would be needed.

We agree with the Reviewer that an adult sample would be an interesting comparison with the current sample. A suggestion for this has been added to the revised manuscript (p. 36). In particular, it would be useful to ensure that the referring situations used in the present study actually invite the interpretation of we both and we all that we presuppose. Nonetheless, we do emphasize that it seems especially unlikely that we both is interpreted as anything other than dyadic by adults (though, this should be empirically examined). Moreover, it seems especially unlikely that we all is interpreted as anything other than a group interpretation (or, perhaps more accurately, a non-dyadic interpretation – or else, why not just say we both)? This latter suggestion might be seen as relying on processes of statistical preemption, as emphasized by Adele Goldberg and others and of which young children are capable (reviewed in Tomasello 2003, Constructing a Language). In any case, we agree that future research should examine

---

## [Decision Letter · Decision Letter 1]

23 Apr 2024

PONE-D-23-24720R1Young English-speaking Children’s Interpretation of Inclusive We: Effects of Group EntitativityPLOS ONE

Dear Dr. Vasil,

Please submit a minor revision that addresses the comments in the review below:

Reviewer #3: Thank you for allowing me the opportunity to read this manuscript. The paper is very clearly and well written, and the methods were obviously very carefully designed. The following “review” is different from a review I normally write. It is instead more like a musing on the current findings (which is a signal that it is an interesting paper!). I do think recruiting some adult participants to engage in the tasks that children did (or some modification of them that is more appropriate for adults) would help disambiguate whether results tell us something about development (only) and/or whether some aspects of study design and linguistic stimuli drive interpretations in some unexpected ways (for all English speakers, regardless of age). The authors suggest doing this in the discussion with lab members, but I would caution against including any adult participants who know anything about the current study and its aims. I motivate this idea in detail in the following:

When I read the abstract for the study and initially scanned the methods, I anticipated three groups to which “we” would refer:

1. The speaker puppet and interlocutor (child); the “dyad” interpretation, included in this paper

2. The speaker puppet and interlocutor (child), along with an/some additional puppet(s); the “group” interpretation, referred to in the paper

3. The speaker puppet and the other puppets; not included in the paper

The interpretation of 3 as a group is not discussed as a possibility in this paper, but when I did my initial reading, I actually thought that that group seemed like the most likely and salient interpretation for “we,” since all three puppets are: 1) on the same side of the computer; 2) in the same physical space; 3) have important attributes in common – they are all animals and puppets! I was worried that the dominance of the puppet group might confuse findings.

Happily, upon reading the methods more carefully, I learned that the researchers designed their methods so that the child always considered themselves a part of the group. Correspondingly, results seem to indicate that the careful design of the study, along with warmup trials and preliminary discourse (e.g., making sure the child had markers, too) encouraged children to consider themselves a member of “we.”

However, I do think the confusing findings related to “we both” may be (at least partially) explained by the salience of the puppet group. “We both” is an odd expression. Since first reading this paper, I’ve been thinking about how I use that phrase. It is harder to think of situations when I would use “we both” to request than it is when I would use it to describe past or future events (“We both felt that way!” or “tomorrow we both are going to go vote”). Importantly, the situations in which I would use “we both” to request/command are ones when I’m emphasizing that another person should do something with me (i.e., I’m emphasizing the “we” by using “both”, e.g., “no, we BOTH need to go vote tomorrow”) rather than clarifying that I mean a dyadic group vs a larger group. To specify a dyad (vs a group), I would say “you and I”. Not “we both”.

Returning to the current paradigm, if I were on a zoom call with a group of people (where the other people were all in the same space, like the puppets are in the current study), and one of them said to me “we both can take notes,” I believe I would interpret “we both” as referring to the speaker and one other member of the group of people in the same physical space, on the other side of the screen. If the speaker wanted to specify that she meant to include me and herself, I would expect her to say, “you and I can take notes” (while looking at me – I see a previous reviewer talked about the role of gaze).

Now, again, I know that the current paradigms do a lot to ensure that the person on the other side of the zoom call (the child in this case) feels like they are always a member of “we”, so this may, again, override the saliency of the fact that the puppets are all in the same room.

Still, this makes me wish that the paradigm was tested on adults. If other adults interpreted sentences the way I (think) I would, they should show a tendency to give markers to two puppets for “we both” and to three puppets for “we all.” Would it be possible to run some version of this paradigm on a few adults, who are naïve to the study’s goals? I do think that their interpretations would be helpful in understanding children’s responses.

Coming back to children’s responses, I suspect that what is described in the discussion relating to quantifiers -- about how young children (don’t) understand quantifiers -- is mostly responsible for current findings, where the addition of “all” or “both” doesn’t reliably predict whether children pick one or more puppets to participate in a task. However, it might be worth taking a quick look at the number of additional puppets selected when they made group interpretations. If participants interpreted “we both” like I think I would (I would guess only four year olds would have this interpretation, if any children did), they should be more likely to select two versus three puppets for “we both” vs “we all.” This effect (if there is one) might be stronger in Study 2, since there are additional factors encouraging children to consider the other puppets as part of the speaker puppet’s group.

For future, I think using an in-person version of this would help as it would eliminate the confounding variable where some people are in one physical space and others are not. I also suggest including unambiguous pronominal phrases like “you and I,” “you,” and “I” to ensure that participants respond to those phrases as expected. This way, when surprising interpretations of “we”, “we both” and/or “we all” are discovered, they can be more assuredly tied to children’s interpretations of these phrases, rather than their confusion about how *any* pronoun works in the current context.

Two additional comments/questions:

The inclusion of sex as a predictor variable is not motivated in the introduction/hypotheses.

What steps were taken to ensure that children were typically developing (or at least typically developing, with respect to language development)?

Please submit the revised manuscript by Jun 07 2024 11:59PM. If you will need more time than this to complete your revisions, please reply to this message or contact the journal office at plosone@plos.org. Please include the following items when submitting your revised manuscript:A rebuttal letter that responds to each point raised by the reviewer. You should upload this letter as a separate file labeled 'Response to Reviewers'.A marked-up copy of your manuscript that highlights changes made to the original version. You should upload this as a separate file labeled 'Revised Manuscript with Track Changes'.An unmarked version of your revised paper without tracked changes. You should upload this as a separate file labeled 'Manuscript'.If applicable, we recommend that you deposit your laboratory protocols in protocols.io to enhance the reproducibility of your results. Protocols.io assigns your protocol its own identifier (DOI) so that it can be cited independently in the future. For instructions see: https://journals.plos.org/plosone/s/submission-guidelines#loc-laboratory-protocols. Additionally, PLOS ONE offers an option for publishing peer-reviewed Lab Protocol articles, which describe protocols hosted on protocols.io. Read more information on sharing protocols at https://plos.org/protocols?utm_medium=editorial-email&utm_source=authorletters&utm_campaign=protocols.

We look forward to receiving your revised manuscript.

Kind regards,

Barbara T Rumain, PhD

Academic Editor

PLOS ONE

Journal Requirements:

Reviewers' comments:

Reviewer's Responses to Questions

**Comments to the Author**

1. If the authors have adequately addressed your comments raised in a previous round of review and you feel that this manuscript is now acceptable for publication, you may indicate that here to bypass the “Comments to the Author” section, enter your conflict of interest statement in the “Confidential to Editor” section, and submit your "Accept" recommendation.

Reviewer #3: (No Response)

2. Is the manuscript technically sound, and do the data support the conclusions?

Reviewer #3: Yes

3. Has the statistical analysis been performed appropriately and rigorously? 

Reviewer #3: I Don't Know

4. Have the authors made all data underlying the findings in their manuscript fully available?

Reviewer #3: Yes

5. Is the manuscript presented in an intelligible fashion and written in standard English?

Reviewer #3: Yes

6. Review Comments to the Author

Reviewer #3: Thank you for allowing me the opportunity to read this manuscript. The paper is very clearly and well written, and the methods were obviously very carefully designed. The following “review” is different from a review I normally write. It is instead more like a musing on the current findings (which is a signal that it is an interesting paper!). I do think recruiting some adult participants to engage in the tasks that children did (or some modification of them that is more appropriate for adults) would help disambiguate whether results tell us something about development (only) and/or whether some aspects of study design and linguistic stimuli drive interpretations in some unexpected ways (for all English speakers, regardless of age). The authors suggest doing this in the discussion with lab members, but I would caution against including any adult participants who know anything about the current study and its aims. I motivate this idea in detail in the following:

When I read the abstract for the study and initially scanned the methods, I anticipated three groups to which “we” would refer:

1. The speaker puppet and interlocutor (child); the “dyad” interpretation, included in this paper

2. The speaker puppet and interlocutor (child), along with an/some additional puppet(s); the “group” interpretation, referred to in the paper

3. The speaker puppet and the other puppets; not included in the paper

The interpretation of 3 as a group is not discussed as a possibility in this paper, but when I did my initial reading, I actually thought that that group seemed like the most likely and salient interpretation for “we,” since all three puppets are: 1) on the same side of the computer; 2) in the same physical space; 3) have important attributes in common – they are all animals and puppets! I was worried that the dominance of the puppet group might confuse findings.

Happily, upon reading the methods more carefully, I learned that the researchers designed their methods so that the child always considered themselves a part of the group. Correspondingly, results seem to indicate that the careful design of the study, along with warmup trials and preliminary discourse (e.g., making sure the child had markers, too) encouraged children to consider themselves a member of “we.”

However, I do think the confusing findings related to “we both” may be (at least partially) explained by the salience of the puppet group. “We both” is an odd expression. Since first reading this paper, I’ve been thinking about how I use that phrase. It is harder to think of situations when I would use “we both” to request than it is when I would use it to describe past or future events (“We both felt that way!” or “tomorrow we both are going to go vote”). Importantly, the situations in which I would use “we both” to request/command are ones when I’m emphasizing that another person should do something with me (i.e., I’m emphasizing the “we” by using “both”, e.g., “no, we BOTH need to go vote tomorrow”) rather than clarifying that I mean a dyadic group vs a larger group. To specify a dyad (vs a group), I would say “you and I”. Not “we both”.

Returning to the current paradigm, if I were on a zoom call with a group of people (where the other people were all in the same space, like the puppets are in the current study), and one of them said to me “we both can take notes,” I believe I would interpret “we both” as referring to the speaker and one other member of the group of people in the same physical space, on the other side of the screen. If the speaker wanted to specify that she meant to include me and herself, I would expect her to say, “you and I can take notes” (while looking at me – I see a previous reviewer talked about the role of gaze).

Now, again, I know that the current paradigms do a lot to ensure that the person on the other side of the zoom call (the child in this case) feels like they are always a member of “we”, so this may, again, override the saliency of the fact that the puppets are all in the same room.

Still, this makes me wish that the paradigm was tested on adults. If other adults interpreted sentences the way I (think) I would, they should show a tendency to give markers to two puppets for “we both” and to three puppets for “we all.” Would it be possible to run some version of this paradigm on a few adults, who are naïve to the study’s goals? I do think that their interpretations would be helpful in understanding children’s responses.

Coming back to children’s responses, I suspect that what is described in the discussion relating to quantifiers -- about how young children (don’t) understand quantifiers -- is mostly responsible for current findings, where the addition of “all” or “both” doesn’t reliably predict whether children pick one or more puppets to participate in a task. However, it might be worth taking a quick look at the number of additional puppets selected when they made group interpretations. If participants interpreted “we both” like I think I would (I would guess only four year olds would have this interpretation, if any children did), they should be more likely to select two versus three puppets for “we both” vs “we all.” This effect (if there is one) might be stronger in Study 2, since there are additional factors encouraging children to consider the other puppets as part of the speaker puppet’s group.

For future, I think using an in-person version of this would help as it would eliminate the confounding variable where some people are in one physical space and others are not. I also suggest including unambiguous pronominal phrases like “you and I,” “you,” and “I” to ensure that participants respond to those phrases as expected. This way, when surprising interpretations of “we”, “we both” and/or “we all” are discovered, they can be more assuredly tied to children’s interpretations of these phrases, rather than their confusion about how *any* pronoun works in the current context.

Two additional comments/questions:

The inclusion of sex as a predictor variable is not motivated in the introduction/hypotheses.

What steps were taken to ensure that children were typically developing (or at least typically developing, with respect to language development)?

7. PLOS authors have the option to publish the peer review history of their article (what does this mean?). If published, this will include your full peer review and any attached files.

Reviewer #3: No

---

## [Author Response · Author response to Decision Letter 1]

10 Jun 2024

Reviewer

We thank the Reviewer for their attention to the manuscript, kind remarks, and helpful comments. We have made minor revisions to the manuscript following their suggestions. Our responses are provided in green text, below, following each comment.

Reviewer #3: Thank you for allowing me the opportunity to read this manuscript. The paper is very clearly and well written, and the methods were obviously very carefully designed. The following “review” is different from a review I normally write. It is instead more like a musing on the current findings (which is a signal that it is an interesting paper!). I do think recruiting some adult participants to engage in the tasks that children did (or some modification of them that is more appropriate for adults) would help disambiguate whether results tell us something about development (only) and/or whether some aspects of study design and linguistic stimuli drive interpretations in some unexpected ways (for all English speakers, regardless of age). The authors suggest doing this in the discussion with lab members, but I would caution against including any adult participants who know anything about the current study and its aims. I motivate this idea in detail in the following:

When I read the abstract for the study and initially scanned the methods, I anticipated three groups to which “we” would refer:

1. The speaker puppet and interlocutor (child); the “dyad” interpretation, included in this paper

2. The speaker puppet and interlocutor (child), along with an/some additional puppet(s); the “group” interpretation, referred to in the paper

3. The speaker puppet and the other puppets; not included in the paper

The interpretation of 3 as a group is not discussed as a possibility in this paper, but when I did my initial reading, I actually thought that that group seemed like the most likely and salient interpretation for “we,” since all three puppets are: 1) on the same side of the computer; 2) in the same physical space; 3) have important attributes in common – they are all animals and puppets! I was worried that the dominance of the puppet group might confuse findings.

Happily, upon reading the methods more carefully, I learned that the researchers designed their methods so that the child always considered themselves a part of the group. Correspondingly, results seem to indicate that the careful design of the study, along with warmup trials and preliminary discourse (e.g., making sure the child had markers, too) encouraged children to consider themselves a member of “we.”

We have revised the Abstract to the manuscript to make our intentions clearer. Specifically, in the revised manuscript we have added text (Abstract) that mentions that “This article reports the results of two online experiments that investigated children’s understanding of inclusive we, in which the child-listener is part of the intended referent of we.” Additionally, we have added text that states that “Participants had their own markers and had to choose to how many partners to distribute (virtual) markers.” 

However, I do think the confusing findings related to “we both” may be (at least partially) explained by the salience of the puppet group. “We both” is an odd expression. Since first reading this paper, I’ve been thinking about how I use that phrase. It is harder to think of situations when I would use “we both” to request than it is when I would use it to describe past or future events (“We both felt that way!” or “tomorrow we both are going to go vote”). Importantly, the situations in which I would use “we both” to request/command are ones when I’m emphasizing that another person should do something with me (i.e., I’m emphasizing the “we” by using “both”, e.g., “no, we BOTH need to go vote tomorrow”) rather than clarifying that I mean a dyadic group vs a larger group. To specify a dyad (vs a group), I would say “you and I”. Not “we both”.

We have revised the Discussion Section to include discussion of this possibility. In the revised Discussion Section (pp. 34-5), we note that “Relatedly, it is plausible that the departure children’s interpretation of we both and we all from predictions does not have to do with their inexperience with these forms. Instead (or in addition), children’s conceptual skills may be involved. Checking this requires additional control conditions that enable comparison of children’s interpretation of we and its quantified forms to that of forms like you and I, or you, me, and Lion. If children’s interpretations of those forms aligns with their interpretations of we both and we all, as in the present studies, then this may suggest that children’s conceptual skills are implicated in explaining the results of the present studies.”

Returning to the current paradigm, if I were on a zoom call with a group of people (where the other people were all in the same space, like the puppets are in the current study), and one of them said to me “we both can take notes,” I believe I would interpret “we both” as referring to the speaker and one other member of the group of people in the same physical space, on the other side of the screen. If the speaker wanted to specify that she meant to include me and herself, I would expect her to say, “you and I can take notes” (while looking at me – I see a previous reviewer talked about the role of gaze).

Now, again, I know that the current paradigms do a lot to ensure that the person on the other side of the zoom call (the child in this case) feels like they are always a member of “we”, so this may, again, override the saliency of the fact that the puppets are all in the same room.

We have revised the Discussion Section to include discussion of this possibility. In the revised Discussion Section (p. 36), we note that “It is possible that the predicted pattern for children is, in fact, not the actual one. A check on this is to conduct an in person analogue of the present studies. This is important because the pragmatics of the current procedure was stilted. Children had to infer that they were part of an intended, first-person plural referent that included individuals who were together, onscreen; children had to infer that they were part of a group with individuals displayed onscreen. This situation is not representative of most children’s naturalistic interactions, which are primarily in person and not online. It will be important, too, to include adults as a control group. This is necessary to ensure that the expected adult interpretations of we both and we all do, in fact, obtain.”

Still, this makes me wish that the paradigm was tested on adults. If other adults interpreted sentences the way I (think) I would, they should show a tendency to give markers to two puppets for “we both” and to three puppets for “we all.” Would it be possible to run some version of this paradigm on a few adults, who are naïve to the study’s goals? I do think that their interpretations would be helpful in understanding children’s responses.

We have revised the Discussion Section to include discussion of the importance of testing adults in future research using the present paradigm. Specifically, in the revised Discussion Section (p. 36), we note that “It will be important, too, to include adults as a control group. This is necessary to ensure that the expected adult interpretations of we both and we all do, in fact, obtain.”

Coming back to children’s responses, I suspect that what is described in the discussion relating to quantifiers -- about how young children (don’t) understand quantifiers -- is mostly responsible for current findings, where the addition of “all” or “both” doesn’t reliably predict whether children pick one or more puppets to participate in a task. However, it might be worth taking a quick look at the number of additional puppets selected when they made group interpretations. If participants interpreted “we both” like I think I would (I would guess only four year olds would have this interpretation, if any children did), they should be more likely to select two versus three puppets for “we both” vs “we all.” This effect (if there is one) might be stronger in Study 2, since there are additional factors encouraging children to consider the other puppets as part of the speaker puppet’s group.

We thank the Reviewer for suggesting these insightful analyses that we had not considered previously. We conducted the analyses suggested. There was some evidence to suggest that the difference in the ratio of 3- to 4-puppet interpretations by 2-year-olds to the same ratio for 4-year-olds was greater in the second study than in the first study. Moreover, this was in the direction predicted by the Reviewer. Specifically, in Study 1, 2-year-olds chose 3-puppet group interpretations 22% of the time (N = 4) and 4-puppet group interpretations 78% of the time (N = 14). The same statistics for 4-year-olds were 26% (N = 7) and 74% (N = 20), respectively. Moreover, in Study 2, 2-year-olds chose 3-puppet group interpretations 17% of the time (N = 5) and 4-puppet group interpretations 83% of the time (N = 25). The same statistics for 4-year-olds were 33% (N = 16) and 67% (N = 32), respectively. However, several models suggested that none of these were reliable statistical differences nor that they varied reliably depending on whether children heard “we both” or “we all.” We have not included discussion of this point in the revised manuscript, but we have retained the code in the associated programming script for interested readers. 

For future, I think using an in-person version of this would help as it would eliminate the confounding variable where some people are in one physical space and others are not. I also suggest including unambiguous pronominal phrases like “you and I,” “you,” and “I” to ensure that participants respond to those phrases as expected. This way, when surprising interpretations of “we”, “we both” and/or “we all” are discovered, they can be more assuredly tied to children’s interpretations of these phrases, rather than their confusion about how *any* pronoun works in the current context.

We kindly point the Reviewer to our second and third comments, above, which respond, respectively, to their comments that (i) “using an in-person version… others are not” and that (ii) “I also suggest… phrases as expected.”

Two additional comments/questions:

The inclusion of sex as a predictor variable is not motivated in the introduction/hypotheses.

We have revised the Introduction Section to better motivate the inclusion of sex as a predictor in the models reported in the studies in the manuscript. Specifically, in the revised Introduction Section (p. 6), we note that “Both studies also investigated effects of sex on interpretation of we. Previous studies of pronoun development suggest that males may form group interpretations of we more often than females (Vasil et al., 2024)”

What steps were taken to ensure that children were typically developing (or at least typically developing, with respect to language development)?

We have revised the Methods Sections of Studies 1 and 2 to clarify the steps taken to ensure that children were typically developing. Our recruitment processes for virtual studies only contact caregivers of participants who are typically developing, as per a statement made by caregivers prior to their child’s participation. We note this in the revised manuscript on pp. 7 and 21 by stating that “Participants… were typically developing (per caregiver statement).

---

## [Editor Report · Decision Letter 2]

20 Jun 2024

Effects of Group Entitativity on Young English-speaking Children’s Interpretation of Inclusive We

PONE-D-23-24720R2

Dear Dr. Vasil,

We’re pleased to inform you that your manuscript has been judged scientifically suitable for publication and will be formally accepted for publication once it meets all outstanding technical requirements.

Kind regards,

Barbara T Rumain, PhD

Academic Editor

PLOS ONE

---

## [Editor Report · Acceptance letter]

27 Jun 2024

PONE-D-23-24720R2 

PLOS ONE

Dear Dr. Vasil, 

I'm pleased to inform you that your manuscript has been deemed suitable for publication in PLOS ONE. Congratulations! Your manuscript is now being handed over to our production team.

Kind regards, 

on behalf of

Dr. Barbara T Rumain 

Academic Editor

PLOS ONE